# Sparse but Wrong: Incorrect L0 Leads to Incorrect Features in Sparse Autoencoders

**David Chanin** [1 2]  **Adrià Garriga-Alonso** [2]

## Abstract

Sparse Autoencoders (SAEs) extract features from LLM internal activations, meant to correspond to interpretable concepts. A core SAE training hyperparameter is L0: how many SAE features should fire per token on average. Existing work compares SAE algorithms using sparsity–reconstruction tradeoff plots, implying L0 is a free parameter with no inherently correct value aside from its effect on reconstruction. In this work we study the effect of L0 on SAEs, and show that if L0 is not set correctly, the SAE fails to disentangle the underlying features of the LLM. If L0 is too low, the SAE will mix correlated features to improve reconstruction. If L0 is too high, the SAE finds degenerate solutions that also mix features. Further, we present a proxy metric that can help guide the search for the correct L0 for an SAE on a given training distribution. We show that our method finds the correct L0 in toy models and coincides with peak sparse probing performance in LLM SAEs. We find that most commonly used SAEs have an L0 that is too low. Our work shows that practitioners must set L0 correctly to train SAEs with monosemantic features.

## 1. Introduction

The Linear Representation Hypothesis (LRH) (Elhage et al., 2022; Park et al., 2024) theorizes that Large Language Models (LLMs) represent concepts as linear directions in representation space. These concepts are nearly orthogonal linear directions, allowing the LLM to represent many more concepts than there are neurons, a phenomenon known as superposition (Elhage et al., 2022). However, superposition poses a challenge for interpretability, as neurons in the LLM are polysemantic, firing on many different concepts.

Sparse autoencoders (SAEs) are meant to reverse superposition, and extract interpretable, monosemantic latent features (Cunningham et al., 2024; Bricken et al., 2023) using sparse dictionary learning (Olshausen & Field, 1997). SAEs have the advantage of being unsupervised, and can be scaled to millions of neurons in its hidden layer (hereafter called "latents"[1]). When training an SAE, practitioners must decide on the sparsity of SAE, measured in terms of L0, or how many latents activate on average for a given input. [2] L0 is typically considered a neutral design choice: most of the literature evaluates SAEs at a range of L0 values, referring to this as a "sparsity–reconstruction tradeoff" (Gao et al., 2024; Rajamanoharan et al., 2024). While most practitioners would expect that too high an L0 will break the SAE due to SAE representations becoming dense, the implication of "sparsity–reconstruction tradeoff" plots is that any sufficiently low L0 is equally valid.

However, recent work shows the same trend: low L0 SAEs perform worse on downstream tasks (Kantamneni et al., 2025; Bussmann et al., 2025). What causes this degraded performance at low L0? In this work, we explore the effect of L0 on SAEs. We begin with toy model experiments using synthetic data, and show that if the L0 is too low, the SAE can "cheat" by mixing together components of correlated features, achieving better reconstruction compared to an SAE with correctly disentangled features. We consider this to be related to feature hedging (Chanin et al., 2025), where the SAE abuses feature correlations to compensate for insufficient resources to model the underlying features monosemantically. This mixing of correlated features into SAE latents affects both positively and negatively correlated features, meaning that in low L0 SAEs, nearly all latents are both less interpretable and more noisy than an SAE with a correctly set L0.

[1]University College London [2]MATS Research. Correspondence to: David Chanin <david.chanin.22@ucl.ac.uk>.

*Proceedings of the $43^{rd}$ International Conference on Machine Learning*, Seoul, South Korea. PMLR 306, 2026. Copyright 2026 by the author(s).

---

[1]We use *latents* to prevent overloading the term *feature*, which we reserve for human-interpretable concepts the SAE may capture. This breaks from earlier usage which used *feature* for both (Elhage et al., 2022), but aligns with the terminology in (Lieberum et al., 2024) and makes the distinction more clear.

[2]TopK and BatchTopK SAEs (Gao et al., 2024; Bussmann et al., 2024) set the L0 ($K$) directly, whereas L1 and JumpReLU (Cunningham et al., 2024; Bricken et al., 2023; Rajamanoharan et al., 2024) adjust it via a coefficient in the loss. In any case, all SAE trainers must decide on the target L0.

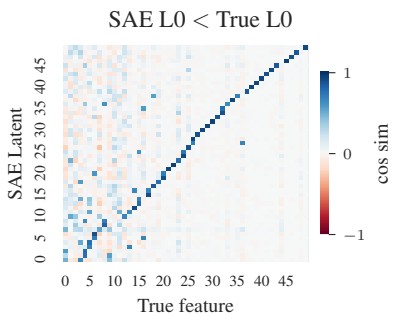 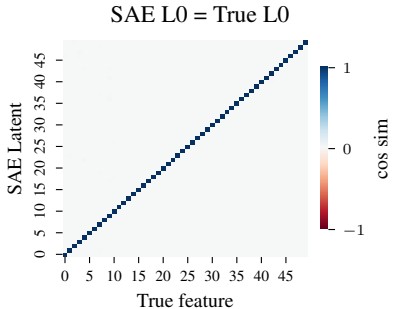 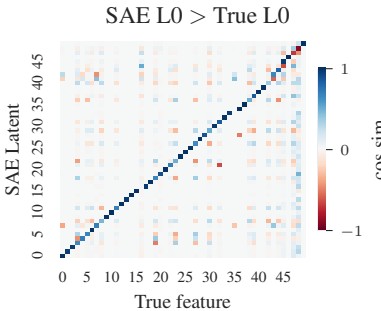

*Figure 1.* When SAE L0 is too low (left) or too high (right), the SAE mixes together correlated features, ruining monosemanticity. Only at the correct L0 (middle), the SAE learns correct features.

Our findings also show that "sparsity–reconstruction trade-off" plots, commonly used to assess SAE architectures, are not a sound method of evaluating SAEs. We demonstrate using toy model experiments that at low L0, an SAE with ground-truth correct latents achieves worse reconstruction than an SAE that mixes correlated features. Thus, if we had an SAE training method that resulted in perfect SAEs, "sparsity–reconstruction tradeoff" plots would cause us to reject that method.

Finally, we develop a proxy metric based on projections between the SAE decoder and training activations that can detect if L0 is too low. We validate these findings on Gemma-2-2b (Team et al., 2024), demonstrating that decoder patterns similar to what we observe in our toy model experiments also manifests in LLM SAEs. We further validate that the optimal L0 we find with our method in Gemma-2-2b matches peak performance on sparse probing tasks (Kantamneni et al., 2025).

Our findings challenge the common belief that any value of L0 is equally valid when training SAEs. This is of direct importance to anyone using SAEs in practice, showing that if L0 is not set correctly, the SAE will become corrupted, learning polysemantic latents that mix features together. Furthermore, our work implies that most SAEs used by researchers today have too low an L0.

Code for experiments is available on Github [3].

## 2. Background

**Sparse autoencoders (SAEs).** An SAE decomposes an input activation $\mathbf{x} \in \mathbb{R}^d$ into a hidden state, $\mathbf{a}$, consisting of $h$ hidden neurons, called "latents". An SAE is composed of an encoder $\mathbf{W}_{\text{enc}} \in \mathbb{R}^{h \times d}$, a decoder $\mathbf{W}_{\text{dec}} \in \mathbb{R}^{d \times h}$, a decoder bias $\mathbf{b}_{\text{dec}} \in \mathbb{R}^d$, and encoder bias $\mathbf{b}_{\text{enc}} \in \mathbb{R}^h$, and a nonlinearity $\sigma$, typically ReLU or a variant like JumpReLU

[3]Code: https://github.com/chanind/sparse-but-wrong-paper

(Rajamanoharan et al., 2024), TopK (Gao et al., 2024) or BatchTopK (Bussmann et al., 2024). The decoder is sometimes called the *dictionary*, in reference to sparse dictionary learning. We use both terms interchangeably.

$$\mathbf{a} = \sigma(\mathbf{W}_{\text{enc}}(\mathbf{x} - \mathbf{b}_{\text{dec}}) + \mathbf{b}_{\text{enc}}) \tag{1}$$
$$\hat{\mathbf{x}} = \mathbf{W}_{\text{dec}}\mathbf{a} + \mathbf{b}_{\text{dec}} \tag{2}$$

In this work we focus on BatchTopK and JumpReLU SAEs as these are both considered SOTA architectures. The JumpReLU activation is a modified ReLU with a threshold parameter $\tau > 0$, so $\text{JumpReLU}_\tau(x) = x \cdot \mathbf{1}_{x > \tau}$. The BatchTopK activation function selects the top $b \times k$ activations across a batch of size $b$, allowing variance in the $k$ selected per sample in the batch. After training, a Batch-TopK SAE is converted to a JumpReLU SAE with a global $\tau$. We follow the JumpReLU training procedure outlined by Anthropic (Conerly et al., 2025).

SAEs are trained as follows, with an auxiliary loss $\mathcal{L}_p$ to revive dead latents with corresponding coefficient $\lambda_p$. JumpReLU SAEs also have a sparsity loss $\mathcal{L}_s$ and corresponding coefficient $\lambda_s$.

$$\mathcal{L} = \|\mathbf{x} - \hat{\mathbf{x}}\|_2^2 + \lambda_s \mathcal{L}_s + \lambda_p \mathcal{L}_p \tag{3}$$

The formulation of $\mathcal{L}_s$ and $\mathcal{L}_p$ for JumpReLU and Batch-TopK SAEs is shown in Appendix B.

## 3. Toy model experiments

The Linear Representation Hypothesis (LRH) (Elhage et al., 2022; Park et al., 2024) states that LLMs represent concepts (alternatively referred to as "features") as (nearly) orthogonal linear directions in representation space. Thus, the hidden activations in an LLM are simply the sum all the firing feature vectors (a feature direction with a positive,

non-zero magnitude) that are being represented. While an LLM can represent a potentially large number of concepts this way, in any given activation, only a small number of concepts are actively represented.

For instance, if we inspect a hidden activation from within an LLM at the token "_Canada", we may expect this activation to be a sum of feature vectors representing concepts like "country", "North America", "starts with C", "noun", etc... The job of a sparse autoencoder is to recover these "true feature" directions in its dictionary.

In a real LLM, we do not have ground-truth knowledge of the "true features" the model is representing, so we do not know if the SAE has learned the correct features. Fortunately, it is easy to create a toy model setup that follows the requirements of the LRH while providing ground-truth knowledge of the underlying true features.

Our toy model has a set of feature embeddings $\mathbf{F} \in \mathbb{R}^{g \times d}$, where $d$ is the input dimension of our SAE, and $g$ is the number of features. All features are orthogonal, so $f_i \cdot f_j = 0$ for $i \neq j$. Each feature $f_i$ fires with probability $p_i$, mean magnitude $\mu_i$, and magnitude standard deviation $\sigma_i$. Feature activations follow a correlated Bernoulli process controlled by correlation matrix $C$, with final magnitudes given by $m_i = a_i \cdot \mathrm{ReLU}(\mu_i + \sigma_i \epsilon_i)$, where $a_i$ indicates whether feature $i$ is active and $\epsilon_i \sim \mathcal{N}(0, 1)$. Training activations for an SAE, $x \in \mathbb{R}^d$, are thus generated as $x = \sum_{i=1}^{n} m_i f_i$

In these toy model experiments, we mainly focus on Batch-TopK SAEs (Bussmann et al., 2024) as this enables direct control of L0. Additionally, we validate our results with JumpReLU SAEs. We train SAEs on 15M synthetic samples with batch size 500 using SAELens (Bloom et al., 2024).

Throughout this section we use the following terminology:

**True L0**   In toy models we have complete control over which features fire, so we know how many features are firing on average. We call this the *true L0* of the model.

**Ground-truth SAE**   Since we know the ground-truth features in our toy models, we can construct an SAE that perfectly captures these features. We refer to this as the *ground-truth SAE*. This is an SAE where $g = h$, $\mathbf{W}_{\mathrm{enc}} = \mathbf{F}^T$, $\mathbf{W}_{\mathrm{dec}} = \mathbf{F}$, $\mathbf{b}_{\mathrm{enc}} = 0$, $\mathbf{b}_{\mathrm{dec}} = 0$.

### 3.1. Low L0 SAEs mix correlated and anti-correlated features

We begin with a small toy model with 5 true features ($g = 5$) in an input space of $d = 20$. We set each $p_i = 0.4$ such that on average 2 features are active per input, for a true L0 of 2. We begin with a simple correlation pattern between features, where $f_0$ is positively correlated with every feature $f_1$ through $f_4$, but otherwise there are no other correlations.

We then train an SAE with $L0 = 2$, matching the true L0 of the model, and an SAE with slightly lower value of $L0 = 1.8$ (BatchTopK SAEs permit setting fractional L0). For the $L0 = 1.8$ SAE, we initialize it to the ground-truth solution, ensuring that the result of training is due to gradient pressure rather than just being a local minimum. We show the toy model feature correlation matrix as well as decoder cosine similarity plots with the true features for both SAEs in Figure 2.

When the SAE L0 matches the true L0, we see that the SAE perfectly learns the underlying true features. However, when SAE L0 is smaller than the true L0, the resulting SAE latents mix feature components together based on the correlation matrix. The latents tracking features $f_1$ through $f_4$ all mix in a *positive* component of $f_0$, but they have no components of each other.

Next, we invert the correlation, i.e. each feature $f_1$ through $f_4$ is negatively correlated with $f_0$ instead, while keeping everything else unchanged. We show the correlation matrix and SAE decoder cosine similarity with true features plots in Figure 3.

Now, we see the same pattern as with positive correlations except inverted. The latents tracking features $f_1$ through $f_4$ mix in a *negative* component of $f_0$, but have no component of each other.

This pattern is problematic because it means that if our L0 is too low, every SAE latent will contain positive components of every positively correlated feature, and negative components of every negatively correlated feature in the model. Negative correlations are particularly bad, as negative correlations are prevalent throughout language. For instance, we may expect a nonsensical negative component of "Harry Potter" to appear in the latent for "French poetry", since Harry Potter has nothing to do with French poetry. This will result in highly polysemantic and noisy SAE latents.

Extended toy model experiments are shown in Appendix D.

### 3.2. Larger toy model experiments

Next, we scale up to a larger toy model with 50 true features ($g = 50$) in input space of $d = 100$. We set $p_0 = 0.345$ and linearly decrease to $p_{49} = 0.05$, so firing probability decreases with feature number. The true L0 of this model is 11. We randomly generate a correlation matrix, so the firings of each feature are correlated with other features. Feature correlations are shown in Appendix C.

We train SAEs with L0 values that are too small ($L0 = 5$), exactly correct ($L0 = 11$), and too large ($L0 = 18$). Results are shown in Figure 1. When the SAE L0 matches the true L0, the SAE exactly learns the true features. When SAE L0 is too low, the SAE mixes components of correlated

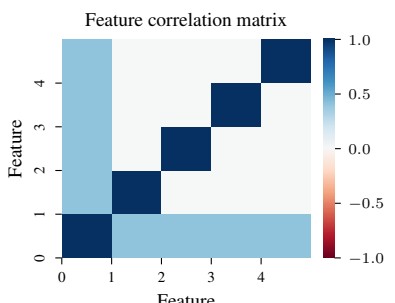
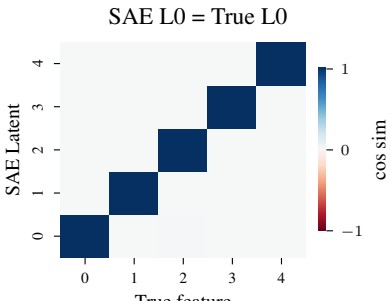
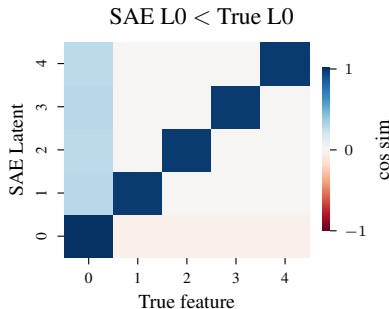

*Figure 2.* (left) Toy model feature correlation matrix showing positive correlations between features. (middle) SAE decoder cosine similarities with true feature when SAE L0 = 2, matching the true L0 of the toy model. (right) SAE decoder cosine similarities with true features when SAE L0 = 1.8. When L0 is too low, the SAE mixes components of features based on their firing correlations.

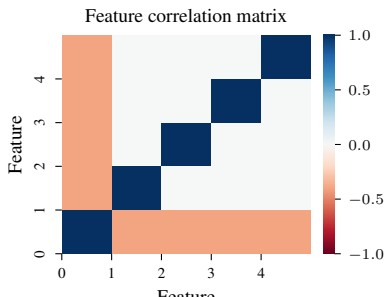
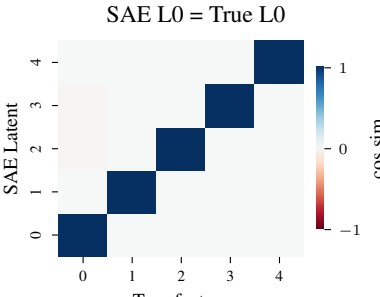
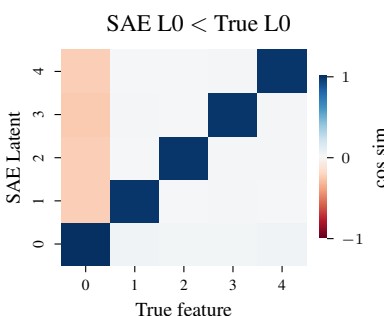

*Figure 3.* (left) Toy model feature correlation matrix showing negative correlations between features. (middle) SAE decoder cosine similarities with true feature when SAE L0 = 2, matching the true L0 of the toy model. (right) SAE decoder cosine similarities with true features when SAE L0 = 1.8. When L0 is too low, the SAE mixes negative components of anti-correlated features.

features together, particularly breaking latents tracking high-frequency features. When L0 is too high, the SAE learns degenerate solutions that mix features together. The further SAE L0 is from the true L0, the worse the SAE. Interestingly, when L0 is too high the SAE still learns many correct latents, but *when L0 is too low, every latent in the SAE is affected*.

### 3.3. MSE loss incentivizes low-L0 SAEs to mix correlated features

Why do SAEs with low L0 not learn the true features? We construct a ground-truth SAE and set $L0 = 5$, to match the low L0 SAE from Figure 1. We generate 100k synthetic training samples and calculate the Mean Square Error (MSE) of both these SAEs. The trained SAE with incorrect latents achieves a MSE of 2.73, while the ground-truth SAE achieves a much worse MSE of 4.88. Thus, *MSE loss actively incentivizes low L0 SAEs to learn incorrect latents*.

This is what our theoretical analysis predicts. Theorem F.1 (Appendix F) proves that when L0 is below the true L0, the disentangled SAE is not the minimizer of expected MSE, and Theorem F.2 strengthens this to show that the disentangled SAE is not even a *local* minimum: there exists a strict descent direction pointing away from it toward mixed

latents. Gradient-based training therefore systematically moves away from the correct dictionary whenever L0 is too low, regardless of initialization.

### 3.4. The sparsity–reconstruction tradeoff

It is common practice to evaluate SAE architectures using a sparsity–reconstruction tradeoff plot (Cunningham et al., 2024; Gao et al., 2024; Rajamanoharan et al., 2024), where the assumption is that having better reconstruction at a given sparsity is inherently better, and indicates that the SAE is correct. After all, we train SAEs to reconstruct inputs, so surely an SAE that has better reconstruction must therefore be a better SAE than one that has lower reconstruction?

Sadly, this is not the case. As we discussed in Section 3.3, when the L0 of the SAE is lower than optimal, the SAE can find ways to "cheat" by engaging in feature hedging (Chanin et al., 2025), and get a better MSE score by mixing components of correlated features together. This results in an SAE where the latents are not monosemantic, and do not track ground-truth features.

We next explore the sparsity–reconstruction tradeoff by training SAEs on our toy model at various L0s. Since we know the ground-truth features in our toy model, we construct a ground-truth SAE that perfectly represents these

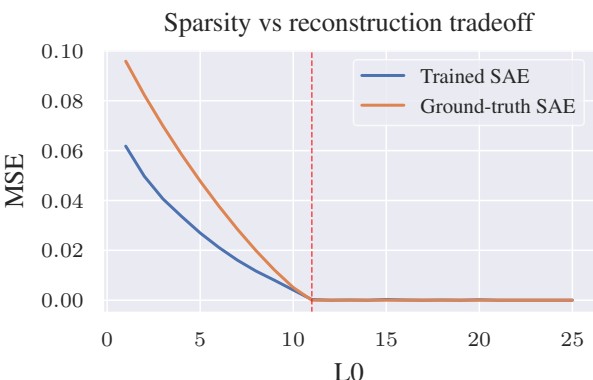

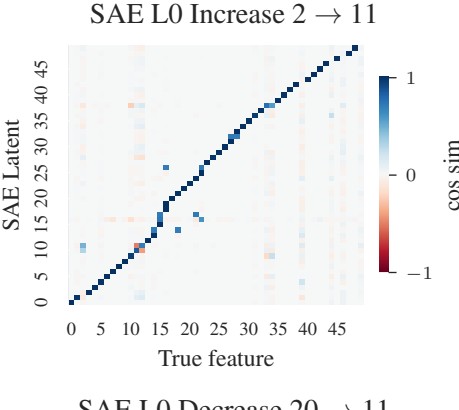

*Figure 4.* Sparsity (*L*0, lower is better) vs reconstruction (mean squared error, lower is better) for learned SAEs and a ground-truth SAE. When L0 is less than the true L0 of the toy model (the dotted line), the trained SAE gets better reconstruction than the ground-truth SAE. Sparsity–reconstruction plots like this lead us to the incorrect conclusion that the ground-truth SAE is a worse SAE.

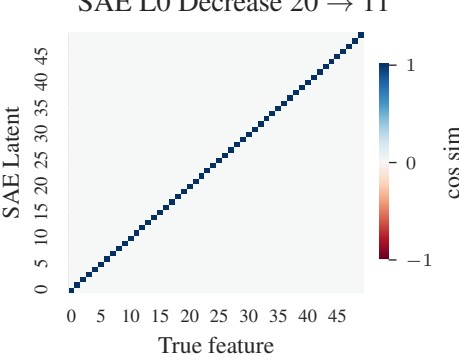

features. We vary the L0 of the ground truth SAE while leaving the encoder and decoder fixed at the correct features. We plot the mean squared error (MSE) vs L0 in Figure 4 for both SAEs. When the SAE L0 is lower than the true L0 of the toy model, the ground-truth SAE scores worse on reconstruction than the trained SAE! If we had an SAE training technique that gave us the ground truth correct SAE for a given LLM, sparsity–reconstruction plots would cause us to discard the correct SAE in favor of an incorrect SAE that mixes features together.

We show the cosine similarity of the SAE decoder latents with the ground truth features for the SAEs learned with L0=1 and L0=2 compared with the ground-truth SAE in Figure 6. Both these SAEs outperform the ground-truth SAE on mean squared error (MSE) despite learning horribly polysemantic latents bearing little resemblance to the underlying true features of the model.

### 3.5. Transitioning L0 during training

We explore the effect of transitioning the L0 of the SAE during training using the toy model from Section 3. This toy model has a true L0 of 11. We train BatchTopK SAEs with a final L0 of 11, but starting with L0 either too high or too low, and linearly transitioning to the correct L0 over the first 25k steps of training, leaving the SAE at the correct L0 for the final 5k steps of training. We use a starting L0 of 20 for the case where we start too high, and use a starting L0 of 2 for the case where we start too low. Results are shown in Figure 5.

We see that decreasing the L0 of the SAE from a too high value to the correct value still results in the SAE learning

*Figure 5.* Transitioning L0 from too low (top) and too high (bottom) to the correct L0 during training. When the starting L0 is too high, the SAE still learns the correct features at the end of training. However, when L0 is too low, the SAE cannot recover fully and still learns many incorrect features at the end of training.

correct features. However, when the SAE starts from a too low L0, the SAE cannot fully recover when the L0 is adjusted to the correct value later. It seems that the latents the SAE learns when L0 is too low is a local minimum that is difficult from the SAE to escape from even when the L0 is later corrected. This is likely because the latents learned when L0 is too low are optimized by gradient descent to achieve a higher MSE loss than is achievable by the correct latents under the same L0 constraint. However, when L0 is too high, there is no equivalent optimization pressure, and is thus less likely to be a local minimum.

### 3.6. Detecting the true L0 using the SAE decoder

Figure 1 reveals that the SAE decoder latents contain mixes of underlying features, both when the L0 is too high and also when it is too low. As the SAE approaches the correct L0, each SAE latent has fewer components of multiple true features mixed in, becoming more monosemantic. Thus, we expect that the closer the SAE is to the correct L0, the more latents should be orthogonal relative to each other, as there are fewer components of shared correlated features mixed into latents. If we are far from the correct L0, then SAE latents contain components of many underlying features,

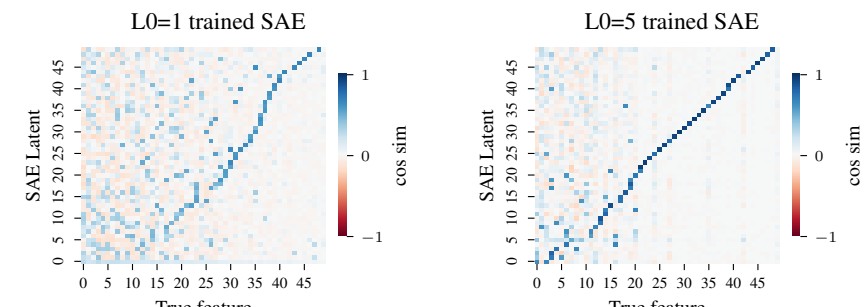

*Figure 6.* SAE decoder cosine similarity with true features for the SAEs from Figure 4 with L0=1 (left) and L0=5 (middle), compared with the ground-truth SAE (right). The trained SAEs score much better than the ground truth SAE on reconstruction, despite their corrupted, polysemantic latents.

and thus we expect latents to have higher cosine similarity with each other.

We call this metric *decoder pairwise cosine similarity*, $c_{\text{dec}}$, and define it as below:

$$c_{\text{dec}} = \frac{1}{\binom{h}{2}} \sum_{i=1}^{h-1} \sum_{j=i+1}^{h} |\cos(\mathbf{W}_{\text{dec},i}, \mathbf{W}_{\text{dec},j})| \quad (4)$$

where $\binom{h}{2} = \frac{h(h-1)}{2}$ is the total number of distinct pairs of latents in the SAE decoder.

If SAE decoder latents are mixing lots of positive and negative components of correlated and anti-correlated features, then each SAE latent should become less orthogonal to each other SAE latent, as many latents will likely mix together similar features. This should mean that the absolute value of the cosine similarity between arbitrary latents should also increase the worse this mixing becomes.

We calculate pairwise calculate similarity $c_{\text{dec}}$ for each of the BatchTopK SAEs we trained on toy models from Section 3.6. Results are shown in Figure 7. We see that pairwise cosine similarity is minimized at the true L0.

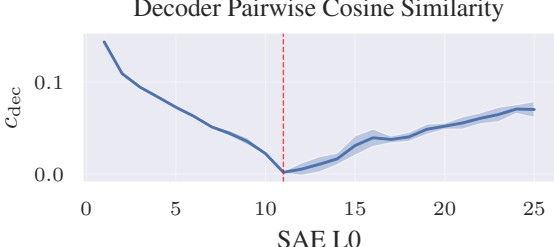

*Figure 7.* Decoder pairwise cosine similarity $c_{\text{dec}}$ evaluated on 5 seeds of toy model SAEs. The true L0 is indicated with a dotted line at 11. Shaded area is 1 stdev. $c_{\text{dec}}$ is minimized at the true L0.

Further toy model experiments exploring the effect superposition and narrow SAE width are shown in Appendix E. Pytorch code implementing $c_{\text{dec}}$ is provided in Appendix M. Formal theoretical justification for the $c_{\text{dec}}$ metric is in Appendix G.

### 3.7. JumpReLU SAE experiments

So far, we have only investigated BatchTopK SAEs due to their ease of setting L0. We now validate that these same conclusions apply to JumpReLU SAEs. We train JumpReLU SAEs with a range of $\lambda_s$ to control the sparsity of the SAEs. We show plots of $\lambda_s$ vs L0 and decoder pairwise cosine similarity vs L0 for these SAEs in Figure 8. We see that the cosine similarity vs L0 broadly follows the same pattern as we saw for BatchTopK SAEs, and is minimized at the correct L0.

Interestingly, we see that the L0 does not change linearly with $\lambda_s$, but instead "sticks" near the correct L0. This is a testament to Anthropic's JumpReLU SAE training method (Conerly et al., 2025), as a wide range of sparsity coefficients $\lambda_s$ cause the SAE to naturally find the correct L0.

## 4. LLM experiments

We train a series of BatchTopK SAEs (Bussmann et al., 2024) with $h = 32768$ on Gemma-2-2b (Team et al., 2024) and Llama-3.2-1b (Dubey et al., 2024) varying L0 and calculate $c_{\text{dec}}$. Each SAE is trained on 500M tokens from the Pile (Gao et al., 2020) using SAELens (Bloom et al., 2024). We also calculate k-sparse probing performance for these SAEs using the benchmark from Kantamneni et al. (2025), consisting of over 100 sparse probing tasks. Results are shown in Figure 9.

The Llama SAE $c_{\text{dec}}$ plot looks very similar to the toy model, with a clear minimum point. The Gemma-2-2b layer 5 SAEs also show a sharp increase in $c_{\text{dec}}$ at low L0 as we saw in toy models, but has a long shallow region with the global

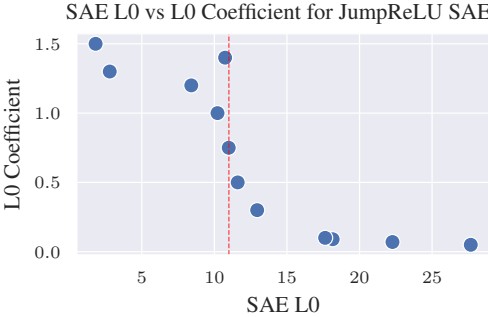

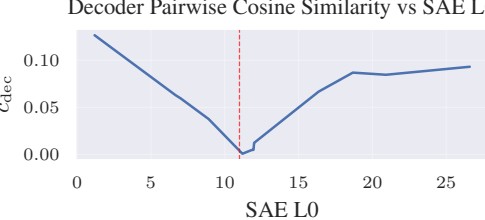

*Figure 8.* (top) L0 coefficient $\lambda_s$ vs L0 for JumpReLU SAEs. (bottom) Decoder pairwise cosine similarity vs L0 for JumpReLU SAEs. The true L0, 11, is marked by a dotted line on the plot.

minimum actually appearing in that shallow region. In both cases, the "elbow" in the $c_{\text{dec}}$ plots just before the jump due to low L0 is around L0 200, and this also corresponds to peak sparse probing performance.

### 4.1. JumpReLU vs BatchTopK SAEs

We next explore how JumpReLU and BatchTopK SAEs compare with decoder pairwise cosine similarity plots. We train a suite of SAEs on 1B tokens on Gemma-2-2b layer 12. We plot $c_{\text{dec}}$ for a range N values as well as k-sparse probing results for JumpReLU and BatchTopK SAEs in Figure 10.

JumpReLU and BatchTopK SAEs behave similarly at low L0, with the high $c_{\text{dec}}$ at low L0 corresponding to poor sparse-probing performance. However, we see notable differences at high L0. The BatchTopK SAEs have a global $c_{\text{dec}}$ minimum around 200, but JumpReLU SAEs $c_{\text{dec}}$ minimum appears closer to 250-300. As in Figure 9, using the "elbow" of the plots just before $c_{\text{dec}}$ jumps due to low L0 seems to correspond to peak k-sparse probing performance.

For JumpReLU SAEs, we see that $c_{\text{dec}}$ rises much less than BatchTopK SAEs at high L0, and indeed, JumpReLU SAEs also perform much better than BatchTopK SAEs at sparse probing when L0 is high. We suspect this is due to JumpReLU SAEs being able to "stick" near the correct threshold per latent like we saw in our toy models section. We investigate the differences in learned SAEs between JumpReLU and BatchTopK further in Appendix L.

### 4.2. Can L0 be both too low and too high simultaneously?

In Figure 10 (right), we plot decoder projection histogram plots for BatchTopK SAEs on Gemma-2-2b layer 12 with L0 10, 200, 750, and 2000. These plots are created by projecting training inputs on the SAE decoder, creating a histogram of how strongly each latents projects onto the input. We expect that the more SAE latents are mixing positive and negative components of underlying features, the more strongly they should project both positively and negatively on arbitrary training inputs. This should look like a narrow Gaussian-shaped histogram around 0 when there is little mixing, and a wider one the more mixing there is. (We use "Gaussian" loosely throughout this section to describe the histogram shape.)

As expected, when L0 is very low (10) or very high (2000), we see a wide Gaussian around 0, indicating that decoder latents are mixing correlated features together. At L0=200, we see a much more narrow distribution around 0, as we expect when near the correct L0. However, at L0=750, we see an interesting phenomenon, where there is an even narrower distribution than at L0=200, but also a large hump starting at projection above 10 (more visible in the log plot).

We suspect this indicates at L0=750, some latents become more monosemantic while other latents mix underlying features becoming less monosemantic. This likely means that the L0 is too high for some latents while simultaneously being too low for other latents. There is no reason why every latent has the same firing threshold, so there is likely a range of L0s where some latents are firing more than they ideally should while other latents are firing less than they ideally should. We also suspect this is part of why JumpReLU SAEs seem to perform much better at high L0, since JumpReLU SAEs can adjust firing threshold per-latent while BatchTopK SAEs cannot.

## 5. Related work

**Limitations of SAEs** Early work on SAEs for interpretability highlight the problem of feature splitting (Bricken et al., 2023; Templeton et al., 2024), where a seemingly interpretable general feature splits into more specific features at narrower SAE widths. Chanin et al. (2025) explores feature hedging, showing SAEs mix correlated features into latents if the SAE is too narrow. We consider our work a version of feature hedging due to low L0. Till (2024) shows SAEs may increase sparsity by inventing features. Chanin et al. (2024) discuss the problem of feature absorption, where SAEs can improve their sparsity score by mixing hierarchical features together. Engels et al. (2024) investigates SAE errors and finds that SAE error may be pathological and non-linear. Engels et al. (2025) find

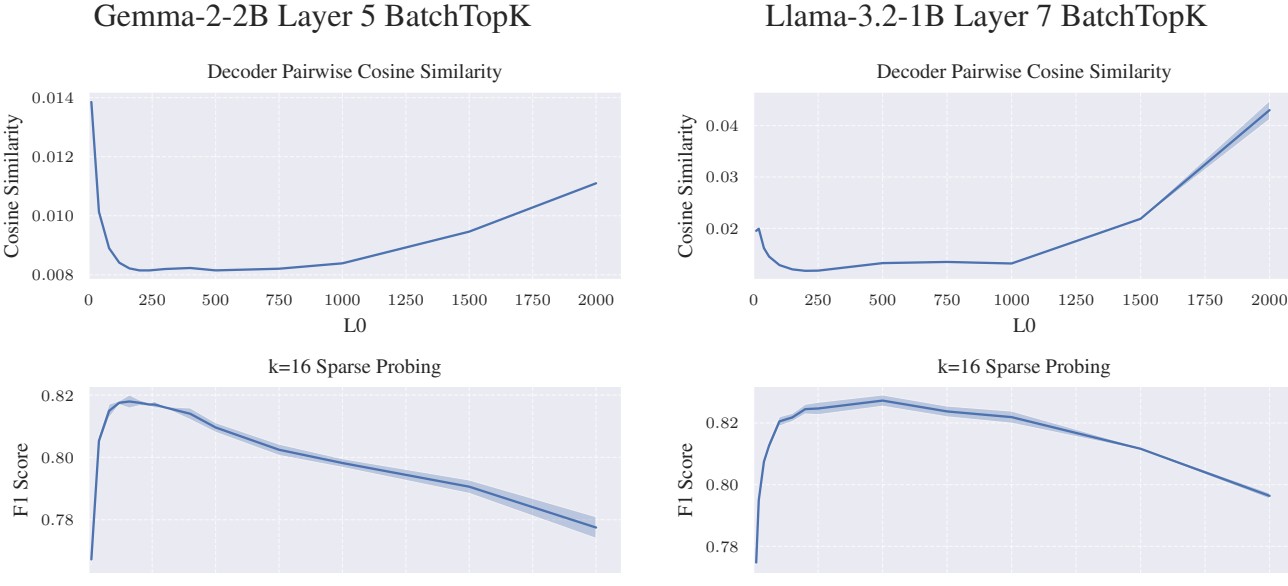

*Figure 9.* Decoder pairwise cosine similarity vs SAE L0 and K-sparse probing F1 vs L0 with 3 seeds per L0. (left) Gemma-2-2b layer 5 BatchTopK results. (right) Llama-3.2-1b layer 7 BatchTopK SAEs. In both cases, peak sparse probing performance occurs in the elbow just before $c_{\text{dec}}$ jumps due to low L0, although the shapes of the $c_{\text{dec}}$ plots vary at high L0.

## Gemma-2-2B Layer 12

*Figure 10.* Gemma-2-2b layer 12, with (left) decoder pairwise cosine similarity and K-sparse probing F1 for BatchTopK and JumpReLU SAEs, and (right) normalized decoder projection histograms for BatchTopK SAEs. Histograms are truncated to -20 and 40 to highlight projections near the origin.

that not all underlying LLM features themselves are linear, demonstrating circular embeddings of some concepts. Wu et al. (2025) and Kantamneni et al. (2025) both investigate empirical SAE performance, finding SAEs underperform relative to supervised baselines, but do not offer theoretical

explanations as to why SAEs underperform.

**Identifiability in dictionary learning**   Related is the question of identifiability in dictionary learning (Gribonval & Schnass, 2010). Wang et al. (2020) prove the true dictio-

nary is the unique sharp local minimum of $\ell_1$-minimization. Spielman et al. (2012) provide polynomial-time algorithms for exact recovery of complete dictionaries. The Donoho-Elad uniqueness theorem (Donoho & Elad, 2003) establishes that a sparse representation is unique only when sparsity is below a threshold determined by the dictionary's mutual coherence, $\mu$ (equivalent superposition in SAEs). Wu & Yu (2018) further show that local identifiability holds when sparsity is $O(\mu^{-2})$ for overcomplete dictionaries. Relatedly, in the regression setting, Zou & Hastie (2005) demonstrate that L1 regularization can exhibit a "grouping effect" assigning them similar coefficients. However, this requires very high correlation (similar to feature absorption). Our work shows that when L0 is too low, even small feature correlations cause mixing.

**Picking SAE hyperparameters**   Related to our work is Minimum Description Lengths (MDL) SAEs (Ayonrinde et al., 2024), which attempt to find reasonable choices for SAE width and L0 based on information theory. However, MDL cannot be used to pick an L0 for a given width, as it requires all SAEs tested to have near-equivalent reconstruction, making it difficult to use in practice. Another SAE architecture which attempts to pick L0 heuristically is Approximate Feature Activation (AFA) SAEs (Lee et al., 2025). AFA SAEs selects L0 adaptively at each input by assuming underlying true features are maximally orthogonal and selecting features until the feature norm is close to the input norm. While the L0 is not set directly in AFA SAEs, there is an extra loss hyperparameter that may modulate the resulting L0, and empirically, seems to result in SAEs with much higher L0 than is typically considered practical.

## 6. Discussion

While most SAE practitioners understand that having too high L0 is problematic, our work shows that having too low L0 is perhaps even worse. Our work has several important implications for the field. First, the L0 used by most SAEs is lower than it should be, as a cursory search of open source SAEs on Neuronpedia (Lin, 2023) shows L0 less than 100 is very common even for SAEs trained on large models (Appendix K). We further show that the sparsity–reconstruction tradeoff, as commonly discussed by most SAE papers (Cunningham et al., 2024; Gao et al., 2024; Rajamanoharan et al., 2024), is misleading: when L0 is too low, an SAE with a correct dictionary achieves worse reconstruction than an incorrect SAE with correlated features.

A further implication concerns SAE-based steering and model editing. If L0 is too low, individual SAE latents represent mixtures of correlated features rather than single concepts, so steering on a single latent activates many underlying features simultaneously, including combinations

that the LLM never sees during training (e.g., a token cannot simultaneously represent a dog, a cat, a bird, and a fish). For anti-correlated features, latents pick up *negative* components of feature directions, and there is no guarantee that the negative of a feature direction corresponds to any valid concept at all. We suspect this can help explain the unpredictable and often disappointing results reported for SAE-based steering (Wu et al., 2025): when the underlying SAE has L0 set too low, steering on its latents pushes the LLM toward directions that simply do not exist in its training distribution.

We presented a metric, $c_{\text{dec}}$, based on the pairwise cosine similarity of the SAE decoder, that can help guide L0 selection. We want to be upfront about its limitations. First, some $c_{\text{dec}}$ curves have broad minima spanning a wide range of L0 (e.g., Figure 9, Gemma-2-2b layer 5), so $c_{\text{dec}}$ is more reliable for ruling out clearly-wrong L0 (especially L0 that is too low) than for pinpointing a precise optimum. Second, while $c_{\text{dec}}$ correlates well with sparse probing performance in our LLM experiments, the correlation is imperfect, particularly at high L0. We therefore recommend treating $c_{\text{dec}}$ as one signal among several when selecting L0, and using it alongside downstream evaluations like sparse probing when those are available. The strongest case for $c_{\text{dec}}$ is in settings where labeled probing benchmarks do not exist (e.g., SAEs trained on biology or chemistry foundation models), where it may be the only practical option for diagnosing whether L0 is set too low.

We hope this investigation into correlation-based SAE quality metrics can be built on further in future work. We are particularly excited about the possibility of learning more about the underlying correlational structure between features by studying correlations in the SAE decoder.

While our metric currently requires training a sweep over L0, in future we hope to optimize L0 automatically during training (steps towards this are discussed in Appendix I).

## Acknowledgments

David Chanin was supported thanks to EPSRC EP/S021566/1 and the Machine Alignment, Transparency & Security (MATS) program. We are grateful to Henning Bartsch and Lovkush Agarwal for feedback during the project.

## Impact Statement

This paper presents work whose goal is to advance the field of Machine Learning. There are many potential societal consequences of our work, none which we feel must be specifically highlighted here.

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

## A. Limitations

We limited the scope of our investigation to features satisfying the linear representation hypothesis, and do not investigate how SAEs react if the underlying features are actually non-linear (Engels et al., 2025). However, we do not feel that non-linear features are necessary for SAEs to fail to work properly, as we demonstrate in this paper. We also do not consider the nuances of how unbalanced correlations impact the SAE, as simple correlations are already enough to cause problems. However, we do expect that different sorts of correlations may affect SAEs differently, and would encourage future work to look into this. We also only investigated a few layers of popular LLMs, as running sweeps of SAE training at every layer of the LLM was too prohibitively expensive for this work. Nevertheless, we have no reason to expect any meaningfully different behavior in decoder projection at other LLM layers.

Finally, we focus our LLM experiments on BatchTopK (Bussmann et al., 2024) and JumpReLU (Rajamanoharan et al., 2024) SAEs, and do not separately evaluate Top-K SAEs (Gao et al., 2024). BatchTopK is architecturally very similar to Top-K, differing only in whether the top-$K$ selection is per-sample or aggregated over a batch, and was introduced as a direct improvement over Top-K. We therefore expect our BatchTopK results to also apply to Top-K SAEs.

## B. SAE training architecture definitions

**Pre-encoder bias.** The $\mathbf{b}_{\text{dec}}$ term subtracted from the input prior to the encoder (see Background) is the same decoder bias that is added back at the decoder output, not a separate parameter. This convention follows Bricken et al. (2023) and recenters inputs around the learned data mean without introducing an additional bias term.

In this work we focus on JumpReLU (Conerly et al., 2025; Rajamanoharan et al., 2024) and BatchTopK (Bussmann et al., 2024) SAEs. For BatchTopK SAEs, there is no sparsity penalty as sparsity is enforced by the BatchTopK function. The auxiliary loss $\mathcal{L}_P$ for BatchTopK is as follows, where $e$ is the SAE training error residual, and $\hat{e}$ is a reconstruction using the top $k_{\text{aux}}$ dead latents (meaning the latents have not fired in more than $n_{\text{dead}}$ steps during training).

$$\mathcal{L}_p = \|e - \hat{e}\|_2^2$$

We follow the JumpReLU training setup from Conerly et al. (2025), which involves both a sparsity loss $\mathcal{L}_s$ and a pre-activation loss for reviving dead latents, $\mathcal{L}_p$. $\mathcal{L}_s$ is defined as below, where $c$ is a scalar scaling factor:

$$\mathcal{L}_s = \sum_i \tanh(c * |a_i| \|\mathbf{W}_{\text{dec},i}\|_2)$$

The pre-activation loss $\mathcal{L}_p$ adds a small penalty for all dead features, where $a_{\text{pre}}$ refers to the pre-activation of the SAE passed into the JumpReLU:

$$\mathcal{L}_p = \sum_i \text{ReLU}(\tau_i - a_{\text{pre},i}) \|\mathbf{W}_{\text{dec},i}\|_2$$

The JumpReLU defines a pseudo-gradient relative to the threshold $\tau$ as follows, where $\epsilon$ is the bandwidth of the estimator:

$$\frac{\partial \text{JumpReLU}(x,\tau)}{\partial \tau} = \begin{cases} -\frac{\tau}{\epsilon} & \text{if } -\frac{1}{2} < \frac{x-\tau}{\epsilon} < \frac{1}{2} \\ 0 & \text{otherwise} \end{cases}$$

## C. Toy model SAE training details

We train on 15M samples with a batch size of 1024 for all toy model experiments, and a learning rate of 3e-4. We do not use any learning rate warm-up or decay. For all SAE latents vs true feature cosine similarity plots, we re-arrange the SAE latents so the latent indices match the feature indices in the plots, as this makes interpreting the plots easier without any loss of generality.

For the large toy model experiments in Section 3.2, we use a randomly generated correlation matrix and linearly decreasing feature firing probabilities, both shown in Figure 11.

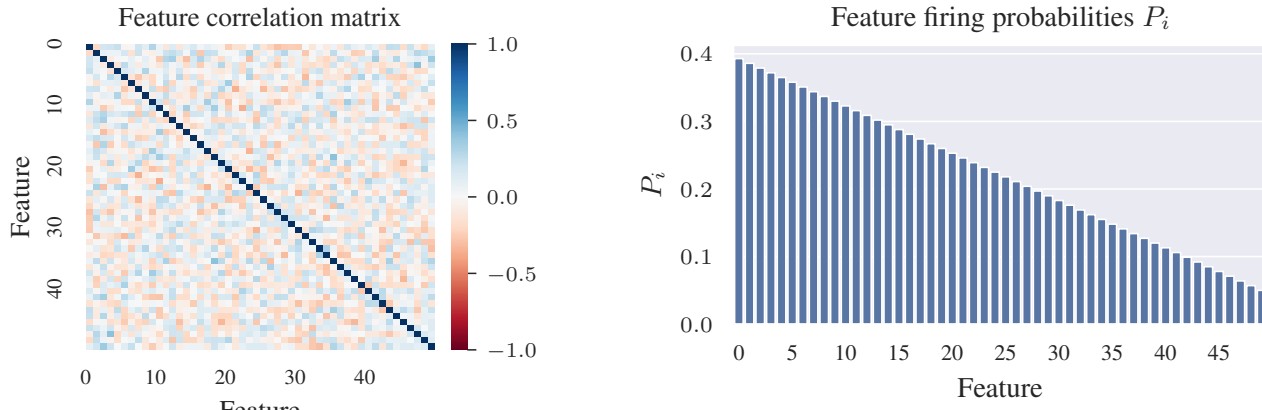

*Figure 11.* (left) random correlation matrix and (right) base feature firing probabilities for toy model.

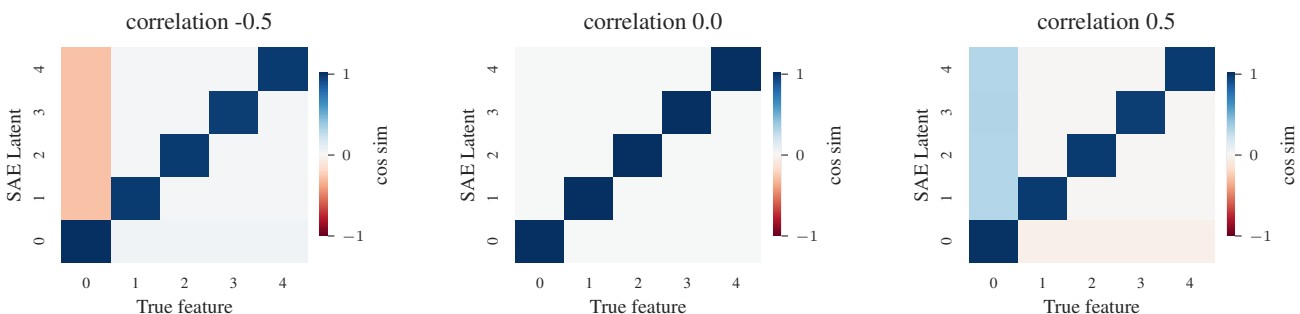

*Figure 12.* Decoder cosine similarity with true features for SAEs trained on toy models with different amounts of correlation between $f_0$ and features 1-4. When there is negative correlation (left), we see the SAE mix in a negative component of $f_0$ into latents 1-4. When there is positive correlation (right), we see a positive component of $f_0$ mixed into the SAE latents. When there is no correlation (middle), there is no mixing.

## D. Extended small toy model experiments

We continue our investigation of feature mixing due to low L0 and correlated features using the same five-feature toy model from Section 3.1.

### D.1. Varying feature correlation strength

We now explore the effect of varying the strength of the correlation between feature $f_0$ and features $f_1$, $f_2$, $f_3$ and $f_4$. In our earlier experiments, we used correlation of $0.4$ and $-0.4$. Here, we will vary correlation between $-0.5$ and $0.5$, while keeping L0=1.8, lower than the true L0 of 2.0. We show decoder cosine similarity plot with true features with correlation $-0.5$, $0.0$, and $0.5$ in Figure 12. As expected, latents 1-4 mix negative components of $f_0$ when correlation is negative, positive components of $f_0$ when correlation is positive, and we see no mixing at all when there is no correlation.

We next measure the amount of mixing by calculating the mean cosine similarity of feature 0 with the SAE latents tracking features $f_1$ through $f_4$. We show results in Figure 13. As expected the more negative the correlation, the more negative the mixing. The more positive the correlation, the more positive the mixing. When there is no correlation, there is no mixing.

### D.2. Varying L0

Next, we study how varying the L0 of the SAE affects the amount of feature mixing we observe. We fix the correlation between feature 0 and features 1-4 at 0.4 for positive correlation, and -0.4 for negative correlation, as in Section 3.1, and vary the L0 of the SAE from 1.7 to 2.0 (the true L0 of the toy model is 2.0). We find that dropping the L0 below 1.7 causes the SAE latents to become so deformed that they bear almost no resemblance to the true features, making it difficult to

Feature mixing vs correlation

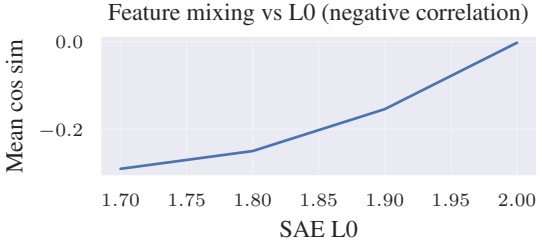

*Figure 13.* Amount of mixing (measured as mean cosine similarity between SAE latents 1-4 and feature 0) vs correlation between feature 0 and features 1-4.

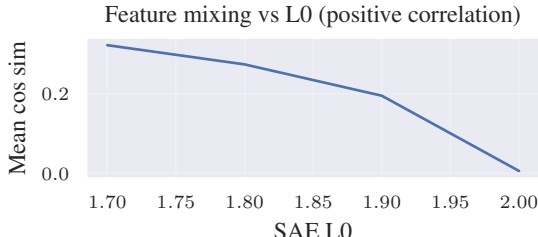

*Figure 14.* Amount of mixing (measured as mean cosine similarity between SAE latents 1-4 and feature 0) vs SAE L0 (true L0 is 2.0). (Left) mixing vs L0 with negative correlation, and (right) mixing vs L0 with positive correlation.

perform systematic analysis.

Results are shown in Figure 14. As expected, the further the SAE L0 gets from the true L0 (2.0), the worse the mixing becomes. Furthermore, the mixing matches the sign of the correlation, with negative correlation causing negative mixing, and positive correlation causing positive mixing.

### D.3. Superposition noise

We have showed that even in the simplest possible setting for an SAE with perfect orthogonality between features, SAEs will fail to learn true features if the SAE L0 is too low and there is correlation between features. If there is superposition noise making the task harder for the SAE, there is thus no reason to expect that SAE will somehow perform better. Regardless, we include results on a toy model with superposition noise below for completeness, as in we use SAEs in situations with superposition noise.

For this experiment, we reuse the same positive and negative correlations from Section 3.1 (+0.4 and -0.4). However, we allow the toy model features to have small positive and negative overlap with each other. We then train an SAE on this toy model with L0=1.9. We find that the using the previous L0=1.8 breaks the SAE too much given the added challenge of superposition noise. We show results in Figure 15.

We see the same pattern as we saw with no superposition noise: the SAE mixes correlated features based on the sign of the correlation. The superposition noise has made the results a bit noisier, but the core trend is still clearly visible.

## E. Extended large toy model experiments

In this section, we build on the results from the 50-latent toy model from Section 3.2.

### E.1. Superposition noise

We now modify the large toy model to have superposition noise, as this is a more realistic setting for an LLM SAE to operate in. We reducing the dimensionality of the space to 40, lower than the number of features in the toy model (50). This forces

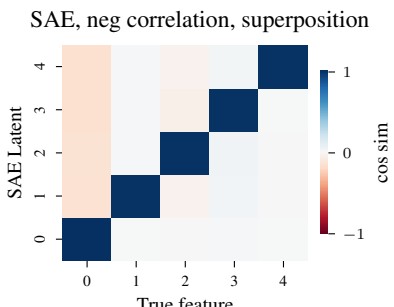
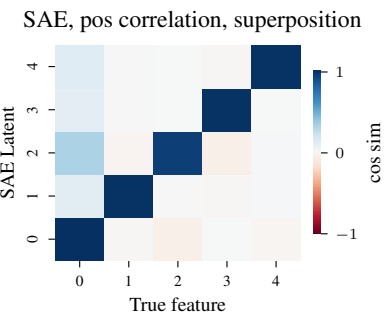

*Figure 15.* Superposition toy model results. For the SAE decoder cosine similarity plots, we subtract out the cosine similarity of the underlying features due to superposition for clarity. (Left) cosine similarity between underlying toy model features, showing positive and negative overlaps between features. (Middle) SAE decoder similarity with true feature with negative correlation between feature 0 and features 1-4. (Right) SAE decoder cosine similarity with true features with positive correlation between feature 0 and features 1-4.

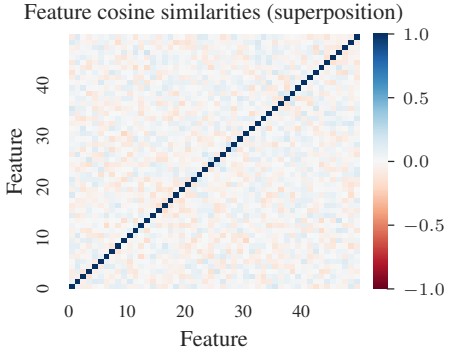
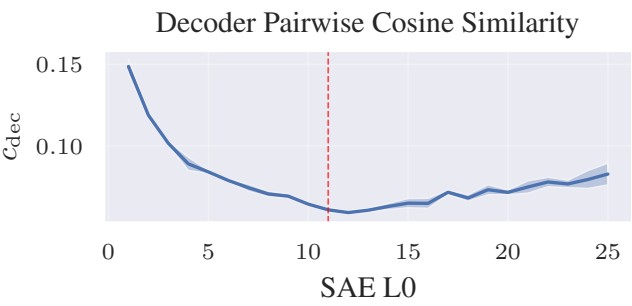

*Figure 16.* (Left) cosine similarity between the true features in the toy model. Due to superposition noise, each features overlaps slightly with many other features. (Right) decoder pairwise cosine similarity for SAEs trained at different L0s. The true L0 is marked with a dashed line.

each feature to slightly overlap other features in the space. The resulting feature cosine similarities are shown in Figure 16 (left).

We train 5 seeds of SAEs at a range of L0s on this superposition toy model, and calculate $c_{\text{dec}}$ in Figure 16 (right). We see that decoder pairwise cosine similarity is still roughly minimized at the true L0 of the toy model.

### E.2. Interaction between SAE width and L0

We mentioned earlier that Chanin et al. (2025) study feature hedging caused by an SAE being too narrow (having fewer latents than there are true features), but how does this interact with mixing due to L0 that we study in this work? To explore this, we train SAEs with 25 latents, so narrower than the 50 features in our toy model.

Our first task is determining the "true L0" in this setting. Our toy model is designed so that earlier features fire with higher probability than later features, so we can be confident that a narrow SAE of width $w$ should preferably learn the first $w$ features of the toy model by index to reduce expected MSE. We thus expect that the "true L0" of our toy model at width $w$ is the L0 of the first $w$ features. For the 25 latent SAEs case we study here, this is a true L0 of 7.6, notably lower than the true L0 of 11 for the full toy model.

Next, we train a suite of narrow BatchTopK SAEs with ranging K from 1 to 20. We plot decoder cosine similarity vs true feature for three of these SAEs in Figure 17. We find that the SAEs near the "true L0" of 7.6 have the most monosemantic latents, while the SAEs with too low or too high L0 are much more corrupted. However, due to feature hedging, even at the "true L0" the SAE still mixes higher index features into its latents, as predicted by the feature hedging work.

Next, we calculate $c_{\text{dec}}$ for these SAEs in Figure 18. As expected, $d_{\text{dec}}$ is minimized at the L0 of the first 25 features in the toy model.

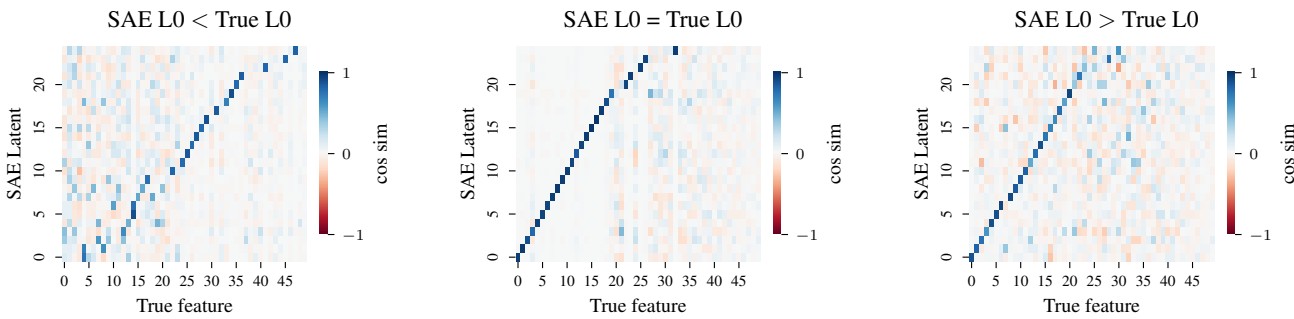

*Figure 17.* SAE decoder cosine similarity with ground-truth for narrow SAEs. When SAE L0 is too low (left) or too high (right), the SAE mixes together correlated features more severely, ruining monosemanticity. At the correct L0 (middle), the SAE features are more monosemantic, but still imperfect due to the feature hedging due SAE width.

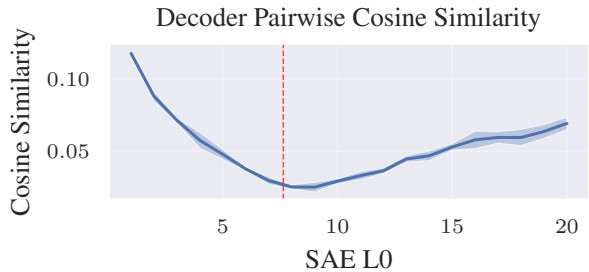

*Figure 18.* Decoder pairwise cosine similarity for narrow SAEs trained at different L0s. The true L0 for the first 25 features is marked with a dashed line. We still see $c_{\text{dec}}$ be minimized at the expected L0.

We see that L0-based feature mixing still occurs as expected in narrow SAEs, the only difference is that we need to adjust the "true L0" expected for the narrow SAEs to account for the fact the that narrow SAEs cannot track as many features as wider SAEs. In general, we should expect the optimal L0 for narrower SAEs to be lower than for for wider SAEs.

## F. Proof: Low L0 incentivizes feature mixing

We now provide a theoretical proof that when SAE L0 is less than the true L0 of the underlying features, MSE loss directly incentivizes the SAE to mix features together.

**Theorem F.1.** *Consider a toy model with two orthonormal features $\mathbf{f}_1, \mathbf{f}_2 \in \mathbb{R}^d$ where $\mathbf{f}_1 \cdot \mathbf{f}_2 = 0$ and $\|\mathbf{f}_1\|_2 = \|\mathbf{f}_2\|_2 = 1$. Let $\mathbf{f}_1$ fire alone with probability $P_1$, $\mathbf{f}_2$ fire alone with probability $P_2$, and both fire together with probability $P_{12}$, where $P_1 + P_2 + P_{12} \leq 1$. Consider a tied SAE with 2 latents (i.e., $\mathbf{W}_{enc} = \mathbf{W}_{dec}^T$) and no biases that can fire at most 1 latent per input (L0 = 1). We assume this is less than the true L0 of the data (i.e., $\mathbb{E}[active\ features] = P_1 + P_2 + 2P_{12} > 1$), which occurs whenever features co-occur ($P_{12} > 0$). Then the SAE that minimizes expected MSE will have latents that mix $\mathbf{f}_1$ and $\mathbf{f}_2$, rather than learning them separately.*

*Proof.* We define our SAE with decoder $\mathbf{W}_{\text{dec}} = [\mathbf{l}_1, \mathbf{l}_2] \in \mathbb{R}^{d \times 2}$ where $\mathbf{l}_1, \mathbf{l}_2$ are the two latent directions. Since the SAE is tied and has no biases, the reconstruction of an input $\mathbf{x}$ using a single active latent $\mathbf{l}_i$ (selected via Top-1 projection) is:

$$\hat{\mathbf{x}} = (\mathbf{l}_i \cdot \mathbf{x})\mathbf{l}_i \tag{5}$$

The reconstruction loss for a single sample is:

$$\mathcal{L}(\mathbf{x}) = \|\mathbf{x} - \hat{\mathbf{x}}\|_2^2 = \|\mathbf{x} - (\mathbf{l}_i \cdot \mathbf{x})\mathbf{l}_i\|_2^2 \tag{6}$$

**Parameterization.** We parameterize the latents as:

$$l_2 = f_2 \tag{7}$$

$$l_1 = \frac{\alpha f_1 + (1 - \alpha) f_2}{\sqrt{\alpha^2 + (1 - \alpha)^2}} \tag{8}$$

where $0 \le \alpha \le 1$ controls the mixture. When $\alpha = 1$, $l_1 = f_1$ (the correct, disentangled solution). When $0 \le \alpha < 1$, $l_1$ mixes both features.

**Case analysis.** We analyze the four possible cases:

**Case 1:** Only $f_1$ fires (probability $P_1$). The input is $x = m_1 f_1$ where $m_1 > 0$ is the magnitude. Latent $l_1$ activates (since it has the largest projection). The reconstruction loss is:

$$\mathcal{L}_1(\alpha) = \|m_1 f_1 - (l_1 \cdot m_1 f_1) l_1\|_2^2 \tag{9}$$

$$= \left\| m_1 f_1 - \frac{m_1 \alpha}{\sqrt{\alpha^2 + (1 - \alpha)^2}} \cdot \frac{\alpha f_1 + (1 - \alpha) f_2}{\sqrt{\alpha^2 + (1 - \alpha)^2}} \right\|_2^2 \tag{10}$$

$$= \left\| m_1 f_1 - \frac{m_1 \alpha^2}{\alpha^2 + (1 - \alpha)^2} f_1 - \frac{m_1 \alpha (1 - \alpha)}{\alpha^2 + (1 - \alpha)^2} f_2 \right\|_2^2 \tag{11}$$

$$= m_1^2 \left[ \left( 1 - \frac{\alpha^2}{\alpha^2 + (1 - \alpha)^2} \right)^2 + \left( \frac{\alpha (1 - \alpha)}{\alpha^2 + (1 - \alpha)^2} \right)^2 \right] \tag{12}$$

Simplifying using $\alpha^2 + (1 - \alpha)^2 = 1 - 2\alpha(1 - \alpha)$:

$$\mathcal{L}_1(\alpha) = m_1^2 \left[ \left( \frac{(1 - \alpha)^2}{\alpha^2 + (1 - \alpha)^2} \right)^2 + \left( \frac{\alpha (1 - \alpha)}{\alpha^2 + (1 - \alpha)^2} \right)^2 \right] \tag{13}$$

$$= m_1^2 \cdot \frac{(1 - \alpha)^2 [\alpha^2 + (1 - \alpha)^2]}{[\alpha^2 + (1 - \alpha)^2]^2} \tag{14}$$

$$= m_1^2 \cdot \frac{(1 - \alpha)^2}{\alpha^2 + (1 - \alpha)^2} \tag{15}$$

**Case 2:** Only $f_2$ fires (probability $P_2$). The input is $x = m_2 f_2$. Latent $l_2 = f_2$ activates, giving perfect reconstruction:

$$\mathcal{L}_2 = 0 \tag{16}$$

**Case 3:** Both $f_1$ and $f_2$ fire (probability $P_{12}$). The input is $x = m_1 f_1 + m_2 f_2$. Since L0 = 1, only one latent can activate. The SAE will choose $l_1$ if $|l_1 \cdot x|^2 > |l_2 \cdot x|^2$. We have:

$$|l_1 \cdot x|^2 = \left( \frac{m_1 \alpha + m_2 (1 - \alpha)}{\sqrt{\alpha^2 + (1 - \alpha)^2}} \right)^2 = \frac{(m_1 \alpha + m_2 (1 - \alpha))^2}{\alpha^2 + (1 - \alpha)^2} \tag{17}$$

$$|l_2 \cdot x|^2 = m_2^2 \tag{18}$$

For simplicity, we assume $m_1 \ge m_2 > 0$, so $l_1$ will activate when $\alpha$ is sufficiently large (e.g., for $\alpha = 1$, $|l_1 \cdot x|^2 = m_1^2 > m_2^2$). Assuming $l_1$ activates, the reconstruction loss is:

$$\mathcal{L}_3(\alpha) = \|m_1 f_1 + m_2 f_2 - (l_1 \cdot (m_1 f_1 + m_2 f_2)) l_1\|_2^2 \tag{19}$$

$$= \left\| m_1 f_1 + m_2 f_2 - \frac{m_1 \alpha + m_2 (1 - \alpha)}{\sqrt{\alpha^2 + (1 - \alpha)^2}} \cdot \frac{\alpha f_1 + (1 - \alpha) f_2}{\sqrt{\alpha^2 + (1 - \alpha)^2}} \right\|_2^2 \tag{20}$$

Let $c = \frac{m_1\alpha + m_2(1-\alpha)}{\alpha^2 + (1-\alpha)^2}$. Then:

$$\mathcal{L}_3(\alpha) = \|m_1\mathbf{f}_1 + m_2\mathbf{f}_2 - c\alpha\mathbf{f}_1 - c(1-\alpha)\mathbf{f}_2\|_2^2 \tag{21}$$

$$= (m_1 - c\alpha)^2 + (m_2 - c(1-\alpha))^2 \tag{22}$$

Expanding and simplifying (see detailed algebra below):

$$\mathcal{L}_3(\alpha) = \frac{[m_1(1-\alpha) - m_2\alpha]^2}{\alpha^2 + (1-\alpha)^2} \tag{23}$$

Note that when $m_1 = m_2 = m$, this simplifies to:

$$\mathcal{L}_3(\alpha) = \frac{m^2(1-2\alpha)^2}{\alpha^2 + (1-\alpha)^2} \tag{24}$$

which equals $0$ when $\alpha = 0.5$ (perfect reconstruction when features are equally mixed) and equals $m^2$ when $\alpha = 1$ (complete failure to reconstruct $\mathbf{f}_2$).

**Case 4:** Neither feature fires (probability $P_0 = 1 - P_1 - P_2 - P_{12}$). Perfect reconstruction with $\mathcal{L}_4 = 0$.

**Expected loss.** The expected loss is:

$$\mathbb{E}[\mathcal{L}(\alpha)] = P_1\mathbb{E}_{m_1}[\mathcal{L}_1(\alpha)] + P_2\mathbb{E}_{m_2}[\mathcal{L}_2(\alpha)] + P_{12}\mathbb{E}_{m_1,m_2}[\mathcal{L}_3(\alpha)] \tag{25}$$

Assuming $\mathbf{l}_1$ activates in Case 1 and $\mathbf{l}_2$ in Case 2 (which holds for $\alpha > 0.5$):

$$\mathbb{E}[\mathcal{L}(\alpha)] = P_1\mathbb{E}_{m_1}\left[m_1^2\frac{(1-\alpha)^2}{\alpha^2 + (1-\alpha)^2}\right] + P_{12}\mathbb{E}_{m_1,m_2}\left[\frac{[m_1(1-\alpha) - m_2\alpha]^2}{\alpha^2 + (1-\alpha)^2}\right] \tag{26}$$

**Concrete example demonstrating feature mixing.** To make this concrete, suppose $m_1 = m_2 = 1$ (both features have equal magnitude when they fire). Assume equal probabilities $P_1 = P_{12} = 0.4$, which implies $P_2 = 0$ (or is negligible) and $P_0 = 0.2$.

For the disentangled solution ($\alpha = 1$, so $\mathbf{l}_1 = \mathbf{f}_1$):

$$\mathcal{L}_1(\alpha = 1) = 0 \quad \text{(perfect reconstruction when only } \mathbf{f}_1 \text{ fires)} \tag{27}$$

$$\mathcal{L}_3(\alpha = 1) = \frac{m^2(1-2)^2}{1^2 + 0^2} = m^2 = 1 \quad \text{(cannot reconstruct } \mathbf{f}_2 \text{ component)} \tag{28}$$

Expected loss: $\mathbb{E}[\mathcal{L}(\alpha = 1)] = (0.4 \times 0) + (0.4 \times 1) = 0.4$

For a mixed solution ($\alpha = 0.6$):

$$\mathcal{L}_1(\alpha = 0.6) = 1^2 \cdot \frac{(1-0.6)^2}{0.6^2 + 0.4^2} = \frac{0.16}{0.52} \approx 0.308 \tag{29}$$

$$\mathcal{L}_3(\alpha = 0.6) = 1^2 \cdot \frac{(1 - 2 \times 0.6)^2}{0.6^2 + 0.4^2} = \frac{(-0.2)^2}{0.52} = \frac{0.04}{0.52} \approx 0.077 \tag{30}$$

Expected loss: $\mathbb{E}[\mathcal{L}(\alpha = 0.6)] = (0.4 \times 0.308) + (0.4 \times 0.077) \approx 0.1232 + 0.0308 = 0.154$

The mixed solution achieves $\mathbb{E}[\mathcal{L}(\alpha = 0.6)] \approx 0.154 < 0.4 = \mathbb{E}[\mathcal{L}(\alpha = 1)]$, demonstrating that MSE loss directly incentivizes feature mixing when L0 is constrained below the true L0.

**Optimal mixing coefficient.** More generally, for the case $m_1 = m_2 = m$, the expected loss is:

$$\mathbb{E}[\mathcal{L}(\alpha)] = \frac{m^2}{\alpha^2 + (1-\alpha)^2} \left[ P_1(1-\alpha)^2 + P_{12}(1-2\alpha)^2 \right] \tag{31}$$

At the boundaries:

- At $\alpha = 1$ (disentangled): $\mathbb{E}[\mathcal{L}(1)] = P_{12}m^2$

- At $\alpha = 0.5$ (maximally mixed): $\mathbb{E}[\mathcal{L}(0.5)] = \frac{P_1 m^2 (0.5)^2}{0.5^2 + 0.5^2} = \frac{P_1 m^2 (0.25)}{0.5} = \frac{P_1 m^2}{2}$

When $P_{12} > P_1/2$, we have $\mathbb{E}[\mathcal{L}(0.5)] < \mathbb{E}[\mathcal{L}(1)]$, showing that mixing reduces loss when both features frequently co-occur. For instance, with $P_1 = P_{12} = 0.5$ and $m = 1$:

$$\mathbb{E}[\mathcal{L}(\alpha = 1)] = 0.5 \tag{32}$$
$$\mathbb{E}[\mathcal{L}(\alpha = 0.5)] = 0.25 \tag{33}$$

This demonstrates that when features frequently co-occur ($P_{12}$ is large), the MSE-optimal solution involves substantial feature mixing ($\alpha^* < 1$) rather than learning them disentangled ($\alpha = 1$), completing the proof. $\qquad\square$

**Detailed algebra for Case 3.** Starting from:

$$\mathcal{L}_3(\alpha) = (m_1 - c\alpha)^2 + (m_2 - c(1-\alpha))^2 \tag{34}$$

where $c = \frac{m_1\alpha + m_2(1-\alpha)}{\alpha^2 + (1-\alpha)^2}$.

Expanding:

$$\mathcal{L}_3 = m_1^2 - 2m_1 c\alpha + c^2\alpha^2 + m_2^2 - 2m_2 c(1-\alpha) + c^2(1-\alpha)^2 \tag{35}$$
$$= m_1^2 + m_2^2 + c^2[\alpha^2 + (1-\alpha)^2] - 2c[m_1\alpha + m_2(1-\alpha)] \tag{36}$$

Note that $c[\alpha^2 + (1-\alpha)^2] = m_1\alpha + m_2(1-\alpha)$ by definition of $c$. Therefore:

$$\mathcal{L}_3 = m_1^2 + m_2^2 + c[m_1\alpha + m_2(1-\alpha)] - 2c[m_1\alpha + m_2(1-\alpha)] \tag{37}$$
$$= m_1^2 + m_2^2 - c[m_1\alpha + m_2(1-\alpha)] \tag{38}$$
$$= m_1^2 + m_2^2 - \frac{[m_1\alpha + m_2(1-\alpha)]^2}{\alpha^2 + (1-\alpha)^2} \tag{39}$$

Further simplification:

$$\mathcal{L}_3 = \frac{(m_1^2 + m_2^2)[\alpha^2 + (1-\alpha)^2] - [m_1\alpha + m_2(1-\alpha)]^2}{\alpha^2 + (1-\alpha)^2} \tag{40}$$

The numerator expands to:

$$(m_1^2 + m_2^2)[\alpha^2 + (1-\alpha)^2] - [m_1^2\alpha^2 + 2m_1 m_2\alpha(1-\alpha) + m_2^2(1-\alpha)^2] \tag{41}$$
$$= m_1^2\alpha^2 + m_1^2(1-\alpha)^2 + m_2^2\alpha^2 + m_2^2(1-\alpha)^2 - m_1^2\alpha^2 - 2m_1 m_2\alpha(1-\alpha) - m_2^2(1-\alpha)^2 \tag{42}$$
$$= m_1^2(1-\alpha)^2 + m_2^2\alpha^2 - 2m_1 m_2\alpha(1-\alpha) \tag{43}$$
$$= [m_1(1-\alpha) - m_2\alpha]^2 \tag{44}$$

We can verify this factorization:

$$[m_1(1-\alpha) - m_2\alpha]^2 = m_1^2(1-\alpha)^2 - 2m_1 m_2\alpha(1-\alpha) + m_2^2\alpha^2 \tag{45}$$

This matches. Therefore:

$$\mathcal{L}_3(\alpha) = \frac{[m_1(1-\alpha) - m_2\alpha]^2}{\alpha^2 + (1-\alpha)^2} \tag{46}$$

**Extension to arbitrary numbers of latents.** Theorem F.1 considered the smallest nontrivial case ($g = 2$, $K = 1$). We now show the same mixing pressure arises whenever $K$ is below the true L0, for any number of features: the disentangled SAE is not even a *local* minimum of expected MSE.

**Theorem F.2.** *Consider $g \geq 2$ orthonormal features $\{\mathbf{f}_1, \ldots, \mathbf{f}_g\} \subset \mathbb{R}^d$. Features activate stochastically, and when active feature $k$ takes magnitude $m_k > 0$. Consider a tied SAE with $g$ latents ($\mathbf{W}_{enc} = \mathbf{W}_{dec}^\top$), no biases, and Top-K activation. Let $T_K$ denote the Top-K set under the disentangled SAE ($\mathbf{l}_k = \mathbf{f}_k$). Suppose there exist indices $i \neq j$ such that*

$$\mathbb{P}(i \in T_K, \ m_j > 0, \ j \notin T_K) > 0, \tag{47}$$

*i.e., with nonzero probability, latent $i$ fires, feature $j$ is active, and latent $j$ is excluded from Top-K. Then the disentangled SAE is not a local minimum of expected MSE.*

Condition (47) holds automatically whenever some input has more than $K$ active features, since then some active feature enters $T_K$ while another is excluded.

*Proof.* Consider the one-parameter family perturbing only latent $i$:

$$\mathbf{l}_i(\varepsilon) = \frac{\mathbf{f}_i + \varepsilon \mathbf{f}_j}{\sqrt{1 + \varepsilon^2}}, \qquad \mathbf{l}_k(\varepsilon) = \mathbf{f}_k \text{ for } k \neq i. \tag{48}$$

At $\varepsilon = 0$ this is the disentangled SAE; all latents remain unit-norm, and only reconstruction in $\mathrm{span}(\mathbf{f}_i, \mathbf{f}_j)$ depends on $\varepsilon$. Assuming the magnitude distribution has no atoms, Top-K ties occur with probability zero, so $T_K$ is locally constant in $\varepsilon$ almost surely and we may differentiate the expected loss through Top-K. We partition the input distribution at $\varepsilon = 0$ into four disjoint events:

$\mathcal{A}$: $i \in T_K, m_j > 0, j \notin T_K$ (the critical event);

$\mathcal{B}$: $i \in T_K, j \in T_K, m_j > 0$;

$\mathcal{C}$: $i \in T_K, m_j = 0$;

$\mathcal{D}$: $i \notin T_K$.

On $\mathcal{D}$ the perturbed latent does not fire, so the contribution is zero. On $\mathcal{A}$, $m_j > 0$ and $j \notin T_K$ force $m_i \geq m_j > 0$ for any $i \in T_K$ by the Top-K ordering, so $m_i m_j > 0$.

**Event $\mathcal{A}$.** Within $\mathrm{span}(\mathbf{f}_i, \mathbf{f}_j)$ the input equals $m_i \mathbf{f}_i + m_j \mathbf{f}_j$. Latent $j$ does not fire ($j \notin T_K$), and latents $k \neq i, j$ have $\mathbf{l}_k = \mathbf{f}_k$ orthogonal to $\mathrm{span}(\mathbf{f}_i, \mathbf{f}_j)$, so only latent $i(\varepsilon)$ contributes to reconstruction in that span. Writing $D = 1 + \varepsilon^2$,

$$\hat{\mathbf{x}}\big|_{\mathrm{span}(\mathbf{f}_i, \mathbf{f}_j)} = (\mathbf{l}_i(\varepsilon)^\top \mathbf{x}) \, \mathbf{l}_i(\varepsilon) = \frac{m_i + \varepsilon m_j}{D}(\mathbf{f}_i + \varepsilon \mathbf{f}_j), \tag{49}$$

yielding residuals $m_i - \hat{x}_{\mathbf{f}_i} = \varepsilon(m_i \varepsilon - m_j)/D$ and $m_j - \hat{x}_{\mathbf{f}_j} = (m_j - \varepsilon m_i)/D$. Noting $(m_j - \varepsilon m_i)^2 = (m_i \varepsilon - m_j)^2$ and summing the squares,

$$E_{\mathcal{A}}(\varepsilon) = \frac{(m_i \varepsilon - m_j)^2(1 + \varepsilon^2)}{D^2} = \frac{(m_i \varepsilon - m_j)^2}{1 + \varepsilon^2}, \qquad \frac{dE_{\mathcal{A}}}{d\varepsilon}\bigg|_{\varepsilon=0} = -2m_i m_j. \tag{50}$$

**Event $\mathcal{B}$.** Both latents fire: latent $j = \mathbf{f}_j$ contributes $m_j \mathbf{f}_j$ and latent $i(\varepsilon)$ contributes as in Event $\mathcal{A}$. The residuals are $\varepsilon(m_i \varepsilon - m_j)/D$ along $\mathbf{f}_i$ and $-\varepsilon(m_i + \varepsilon m_j)/D$ along $\mathbf{f}_j$; a short calculation gives $E_{\mathcal{B}}(\varepsilon) = \varepsilon^2(m_i^2 + m_j^2)/(1 + \varepsilon^2)$, which is $O(\varepsilon^2)$.

**Event $\mathcal{C}$.** Setting $m_j = 0$ in the Event $\mathcal{A}$ derivation gives $E_{\mathcal{C}}(\varepsilon) = m_i^2 \varepsilon^2/(1 + \varepsilon^2)$, also $O(\varepsilon^2)$. Hence

$$\frac{dE_{\mathcal{B}}}{d\varepsilon}\bigg|_{\varepsilon=0} = \frac{dE_{\mathcal{C}}}{d\varepsilon}\bigg|_{\varepsilon=0} = 0. \tag{51}$$

**Combining.** By (47), $\mathbb{P}(\mathcal{A}) > 0$ and $m_i m_j > 0$ on $\mathcal{A}$, so

$$\left.\frac{d\mathcal{L}}{d\varepsilon}\right|_{\varepsilon=0} = -2\,\mathbb{P}(\mathcal{A})\,\mathbb{E}[m_i m_j \mid \mathcal{A}] < 0. \tag{52}$$

Thus the disentangled SAE is not a local minimum of expected MSE. $\square$

*Remark* F.3. The same argument applies to every pair $(i, j)$ satisfying (47), so when $K$ is below the true L0 the descent pressure acts on nearly every latent simultaneously, consistent with the empirical observation in Figure 1 that every latent is affected when L0 is too low. Only co-occurrence on Event $\mathcal{A}$ is required, not statistical correlation; correlation determines the magnitude and sign of $\mathbb{E}[m_i m_j \mid \mathcal{A}]$, and for anti-correlated features the descent direction is $-\varepsilon$, matching the negative mixing coefficients observed in Section 3.

## G. Theoretical Justification for $c_{\mathrm{dec}}$ Metric

We provide a theoretical justification for why the decoder pairwise cosine similarity metric, $c_{\mathrm{dec}}$, serves as a proxy for detecting feature mixing in SAEs.

**Theorem G.1.** *Consider two SAEs with identical dictionary size $h$, where SAE 1 learns disentangled features and SAE 2 mixes a correlated feature into its latents. Let the underlying true features $\{\mathbf{f}_1, \ldots, \mathbf{f}_h, \mathbf{g}\}$ be an orthonormal set in $\mathbb{R}^d$, where $\mathbf{f}_i$ are unique features and $\mathbf{g}$ is a dense or frequent feature correlated with multiple $\mathbf{f}_i$. We model the decoder weights $\mathbf{W}_{dec,i}$ (normalized to unit length) for the two SAEs as:*

$$\text{SAE 1 (Disentangled):} \quad \mathbf{W}_i^{(1)} = \mathbf{f}_i \tag{53}$$

$$\text{SAE 2 (Mixed):} \quad \mathbf{W}_i^{(2)} = \sqrt{1 - \gamma_i^2}\,\mathbf{f}_i + \gamma_i \mathbf{g} \tag{54}$$

*where $\gamma_i \in [-1, 1]$ represents the mixing coefficient for latent $i$. Assume there exists a subset of latents $S \subseteq \{1, \ldots, h\}$ with $|S| \geq 2$ such that for all $i \in S$, $\gamma_i \neq 0$. Then, the expected pairwise cosine similarity is strictly greater for SAE 2 than SAE 1:*

$$c_{dec}(\text{SAE 2}) > c_{dec}(\text{SAE 1}) \tag{55}$$

*Proof.* Recall the definition of decoder pairwise cosine similarity:

$$c_{\mathrm{dec}} = \frac{1}{\binom{h}{2}} \sum_{i=1}^{h-1} \sum_{j=i+1}^{h} |\cos(\mathbf{W}_{\mathrm{dec},i}, \mathbf{W}_{\mathrm{dec},j})| \tag{56}$$

Since the decoder weights are normalized, $\cos(\mathbf{W}_{\mathrm{dec},i}, \mathbf{W}_{\mathrm{dec},j}) = \mathbf{W}_{\mathrm{dec},i}^\top \mathbf{W}_{\mathrm{dec},j}$.

**Case 1: SAE 1 (Disentangled).** For any distinct pair $i \neq j$, the weights are $\mathbf{W}_i^{(1)} = \mathbf{f}_i$ and $\mathbf{W}_j^{(1)} = \mathbf{f}_j$. Since the underlying features are orthonormal:

$$\mathbf{W}_i^{(1)\top} \mathbf{W}_j^{(1)} = \mathbf{f}_i^\top \mathbf{f}_j = 0 \tag{57}$$

Thus, for SAE 1:

$$c_{\mathrm{dec}}(\text{SAE 1}) = 0 \tag{58}$$

**Case 2: SAE 2 (Mixed).** Consider the dot product for a distinct pair $i, j$:

$$\mathbf{W}_i^{(2)\top} \mathbf{W}_j^{(2)} = (\sqrt{1 - \gamma_i^2}\,\mathbf{f}_i + \gamma_i \mathbf{g})^\top (\sqrt{1 - \gamma_j^2}\,\mathbf{f}_j + \gamma_j \mathbf{g}) \tag{59}$$

$$= \sqrt{(1 - \gamma_i^2)(1 - \gamma_j^2)}(\mathbf{f}_i^\top \mathbf{f}_j) + \gamma_j \sqrt{1 - \gamma_i^2}(\mathbf{f}_i^\top \mathbf{g})$$

$$+ \gamma_i \sqrt{1 - \gamma_j^2}(\mathbf{g}^\top \mathbf{f}_j) + \gamma_i \gamma_j (\mathbf{g}^\top \mathbf{g}) \tag{60}$$

Using the orthonormality of the set $\{\mathbf{f}_1, \ldots, \mathbf{f}_h, \mathbf{g}\}$:

- $\mathbf{f}_i^\top \mathbf{f}_j = 0$

- $\mathbf{f}_i^\top \mathbf{g} = 0$ and $\mathbf{g}^\top \mathbf{f}_j = 0$

- $\mathbf{g}^\top \mathbf{g} = 1$

The expression simplifies to:

$$\cos(\mathbf{W}_i^{(2)}, \mathbf{W}_j^{(2)}) = \gamma_i \gamma_j \tag{61}$$

The metric $c_{\text{dec}}$ is the average of absolute cosine similarities:

$$c_{\text{dec}}(\text{SAE 2}) = \frac{1}{\binom{h}{2}} \sum_{i<j} |\gamma_i \gamma_j| \tag{62}$$

Since we assumed there exists a subset $S$ where $\gamma_i \neq 0$, there exists at least one pair $(i, j)$ where $|\gamma_i \gamma_j| > 0$. All other terms are non-negative. Therefore:

$$c_{\text{dec}}(\text{SAE 2}) > 0 = c_{\text{dec}}(\text{SAE 1}) \tag{63}$$

This confirms that mixing a shared feature component into multiple latents strictly increases the $c_{\text{dec}}$ metric.  □

*Remark* G.2. In real-world scenarios with superposition noise, the baseline orthogonality $\mathbf{f}_i^\top \mathbf{f}_j$ is not exactly zero but follows a distribution with mean zero and variance $\approx 1/d$. However, systematic feature mixing introduces a structured non-zero component ($\gamma_i \gamma_j$) that typically dominates the random superposition noise, causing a measurable rise in $c_{\text{dec}}$ as observed in Figure 7 and Figure 9.

# H. LLM SAE training details

For BatchTopK SAEs, we ensure that the decoder remains normalized with $||\mathbf{W}_{\text{dec}}||_2 = 1$. We use a learning rate of $3e^{-4}$ with no warmup or decay.

For JumpReLU SAEs, we broadly follow the training procedure laid out by Conerly et al. (2025). However, we do not apply learning rate decay, and only warm $\lambda_s$ for 100M tokens to avoid the sparsity penalty changing throughout the majority of training. We use a learning rate of $2e^{-4}$, $c = 4$, $\lambda_p = 3e^{-6}$ and bandwidth $\epsilon = 2.0$ as recommended by (Conerly et al., 2025).

# I. Automatically finding the correct L0 during training

A natural next step of our finding that the correct L0 occurs when the decoder pairwise cosine similarity, $c_{\text{dec}}$, is minimized is to use this to find the correct L0 automatically during training. This is a meta-learning task, as the L0 is a hyperparameter of the training process. We find there are several challenges to directly using $c_{\text{dec}}$ as an optimization target:

- **Small gradients directly above correct L0** In our plots of $c_{\text{dec}}$, we find that the metric is relatively flat in a region start at the correct L0 and extending to higher L0 values. We thus need a way to traverse this flat region and stop once the metric starts to increase again.

- **The impact of changing L0 is delayed** We find that it takes many steps after changing L0 for $c_{\text{dec}}$ to also change, meaning it is easy to overshoot the target L0 or oscillate back and forth.

- **Dropping L0 too low can harm the SAE** As we saw in Section 3.5, if the L0 is too low the SAE can permanently end up in poor local minima. We thus want to avoid dropping below the correct L0, even temporarily, to avoid permanently breaking the SAE. We therefore need to start with L0 too high and slowly decrease it until we find the correct L0.

- **Noise during training** We find that while $c_{\text{dec}}$ shows clear trends after training for many steps, it can be noisy on each training sample. So our optimization needs to be robust to this noise.

Taking these requirements into account, we present an optimization procedure to find the L0 that minimizes $c_{\text{dec}}$ automatically during training. We first estimate the gradient of $c_{\text{dec}}$, hereafter referred to as to as the metric, $m$, with respect to L0, $dm/dL0$.

We first define an evaluation step $t$ as a set number of training steps (we evaluate every 100 training steps). At $t$ we change L0 by $\delta_{L_0}$. At the next evaluation step, $t + 1$, we evaluate $m$. We use a sliding average of $c_{\text{dec}}$ over the past 10 training steps to calculate $m$ to help account for noise. We the estimate $dm/dL0$ as:

$$\frac{dm}{dL0} = \frac{m_{t+1} - m_t}{\delta_{L0}}$$

Next, we add a small negative bias to this gradient estimate to encourage our estimate to push L0 lower even if the loss landscape is relatively flat. We use a bias magnitude $0 < b < 1$ that is multiplied by the magnitude of our gradient estimate, so that our biased estimate can never change the sign of the gradient estimate, but can gently nudge it to be more negative in flat, noisy regions of the loss landscape. We find $b = 0.1$ works well. Thus, our biased gradient estimate $dm_b/dL0$ is calculated as below:

$$\frac{dm_b}{dL0} = \frac{dm}{dL0} - b \left| \frac{dm}{dL0} \right|$$

We then provide this gradient to the Adam optimizer ([Kingma & Ba, 2014](#)) with default settings, and allow it to change the L0 parameter.

We add the following optional modifications to this algorithm. First, we clip the gradient estimates $dm/dL0$ to be between -1 and 1. We also set a minimum and maximum $\delta_{L_0}$. The minimum is added to avoid the denominator of our gradient estimate being near 0, and the maximum is chosen to keep the L0 from changing too quickly. In practice, we find a minimum $\delta_{L_0}$ between 0 and 1 seems to work well, and a maximum $\delta_{L_0}$ between 1 and 5 seems to work well.

We find that this optimization strategy works very well in toy models, but requires a lot of hyperparameter tuning to work in real LLMs, limiting its utility. The starting L0, $b$, learning rate for the Adam optimizer, and min and max $\delta_{L_0}$ values all have a big impact on how fast and how aggressively the optimization works. The slope of $m$ around the correct L0 is shallow, so it is easy to overshoot. We expect it is possible to further simplify and improve this process in future work.

## J. Extended LLM results

We include further results for Gemma-2-2b layer 20, to extend the analysis to later model layers. Results are shown in Figure 19.

## K. L0 of open-source SAEs on Neuronpedia

We analyze common open-source SAEs as provided by Neuronpedia ([Lin, 2023](#)) and SAELens ([Bloom et al., 2024](#)). We include all SAEs cross-listed in both SAELens and Neuronpedia with an L0 reported in SAELens. We show the results as a histogram in Figure 20. Our analysis shows that for layer 12 of Gemma-2-2b, the correct L0 should be around 200-250. However, we find that most open-source SAEs have L0 below 100, much lower than our analysis expects to be ideal.

## L. Extended analysis of JumpReLU vs BatchTopK dynamics

JumpReLU and BatchTopK SAEs are both considered state of the art, but we find they have notable differences in their behavior at high L0 in our experiments. In this section, we explore what maybe be causing these differences. In theory, JumpReLU and BatchTopK SAEs are very similar, as a BatchTopK SAE can be viewed as a JumpReLU SAE with a single global threshold, rather than a threshold per-latent ([Bussmann et al., 2024](#)). However, the training losses are quite different for JumpReLU vs BatchTopK. We use the JumpReLU variant laid out by [Conerly et al. (2025)](#), which allows gradients to flow through the JumpReLU threshold to the rest of the model parameters. We expect this means that JumpReLU SAEs are better able to coordinate the threshold with the rest of the model parameters, while BatchTopK cannot, as the threshold does not directly receive a gradient in BatchTopK training.

We begin by comparing the encoder bias between JumpReLU and BatchTopK in Figure 21. We see that BatchTopK SAEs rely much more heavily on the encoder bias than JumpReLU SAEs seem to, with a much wider variance in values and a sharper decrease compared to JumpReLU. We expect this is because BatchTopK cannot coordinate the cutoff threshold with

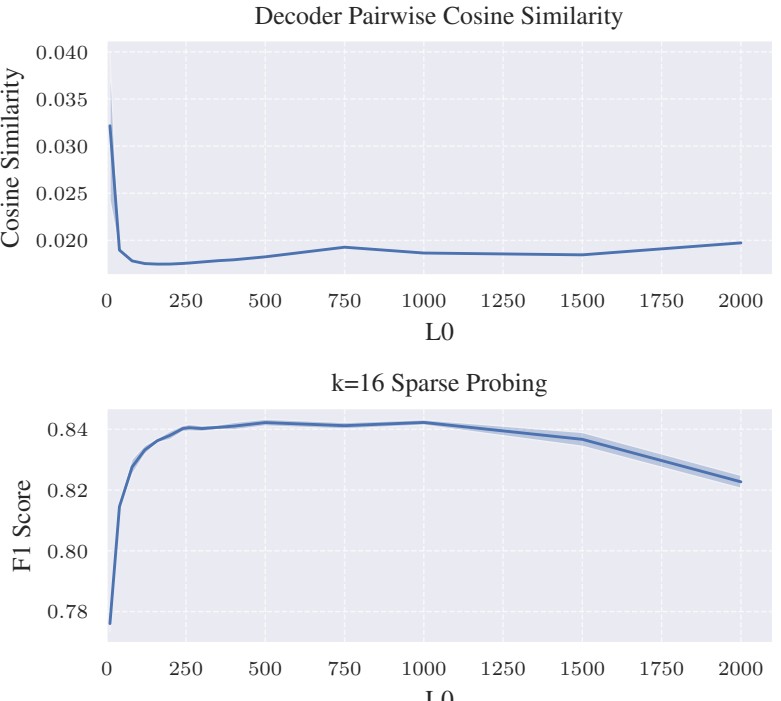

*Figure 19.* Decoder pairwise cosine similarity $c_{dec}$ (top) and K=16 sparse probing results (bottom) for BatchTopK SAEs trained on Gemma-2-2b layer 20.

the encoder directly as JumpReLU can, since there is no gradient available to directly change the threshold of BatchTopK SAEs.

Next, we inspect the threshold values between JumpReLU and BatchTopK in Figure 22. Here as well, we see dramatic differences between BatchTopK and JumpReLU SAEs. The threshold for BatchTopK is much higher than it is for JumpReLU, and the threshold decreases as L0 increses. This makes sense, since using a lower cutoff means more latents can fire. However, JumpReLU seems to unintuitively have the opposite trend, with the threshold actually *increasing* with L0. We saw in Figure 21 that the encoder bias for JumpReLU (and BatchTopK) SAEs increases as well as L0 increases, so perhaps this increase in threshold for JumpReLU SAEs with increasing L0 is just to offset that trend somewhat. We also notice that the variance in JumpReLU SAE thresholds also increases as L0 increases, supporting our hypothesis that one of the reasons JumpReLU SAEs seem to handle high L0 better than BatchTopK is because the thresholds are able to dynamically adjust to near the correct cutoff point per latent, alleviating the situation we saw in BatchTopK SAEs where we can be at both too high and too low L0 at the same time (Section 4.2).

## M. Pytorch pseudocode for $c_{dec}$

We present Pytorch pseudocode for decoder pairwise cosine similarity in Figure 23.

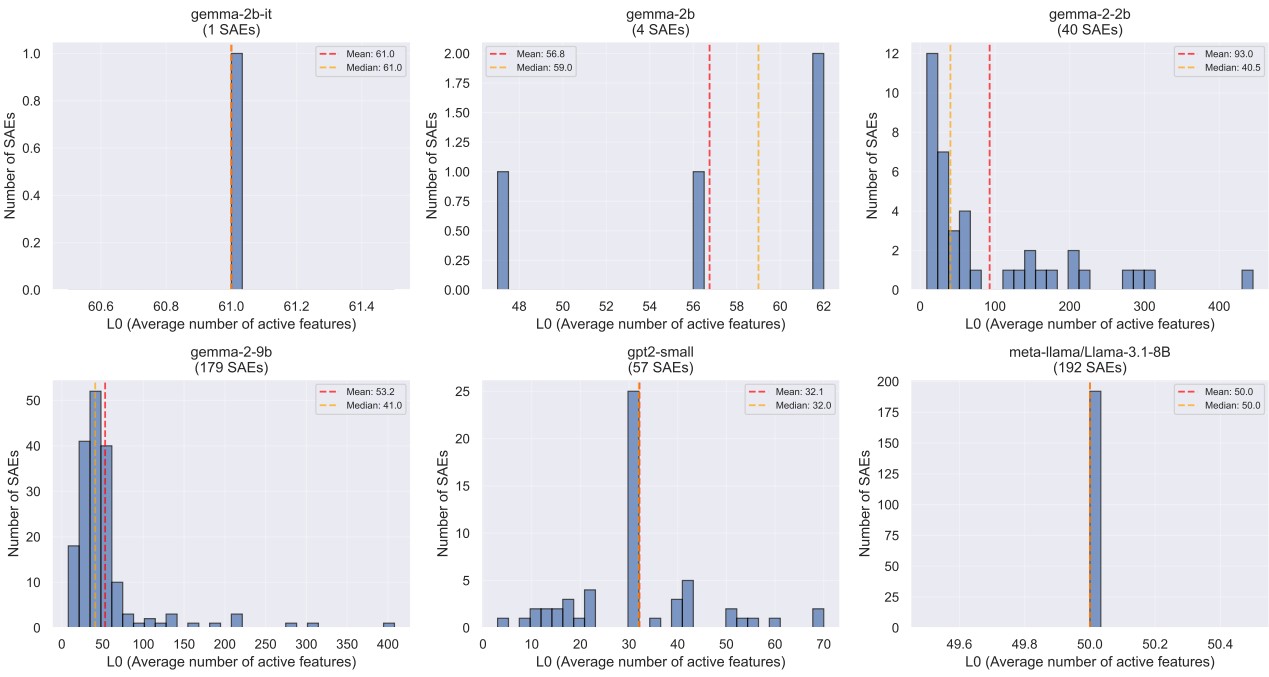

*Figure 20.* L0 of SAEs on Neuronpedia with known L0 listed in SAELens.

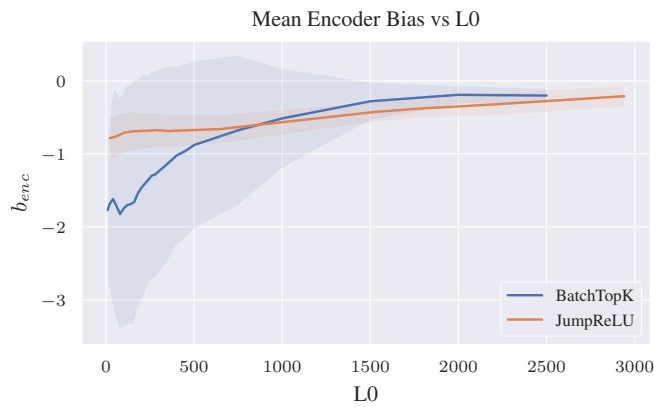

*Figure 21.* Mean encoder bias vs L0. Shaded area in plots corresponds to 1 stdev.

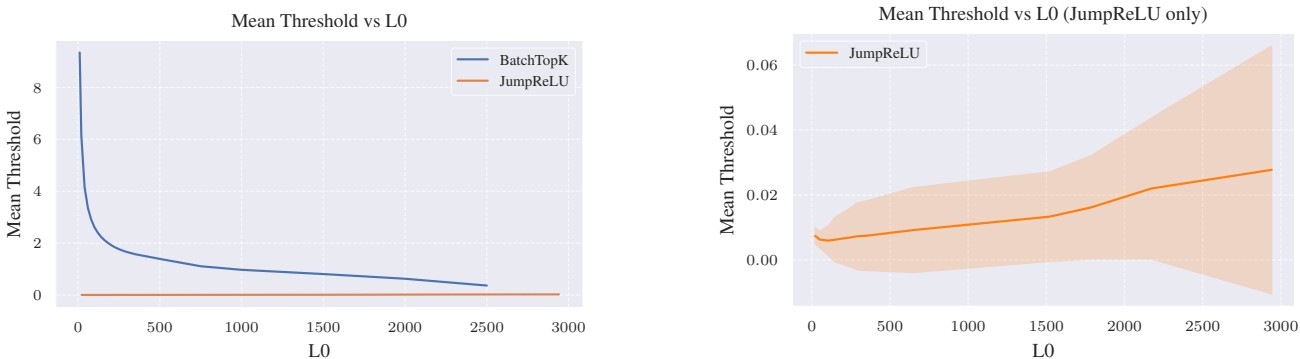

*Figure 22.* Threshold vs L0 for JumpReLU and BatchTopK SAEs. Shaded area in plots corresponds to 1 stdev. Interestingly, JumpReLU threshold is much lower than the BatchTopK threshold, and actually increases as L0 increases. We plot just JumpReLU on its own (right) since it is otherwise difficult to see these trends, as the threshold is so much smaller than BatchTopK.

```python
def pairwise_decoder_cosine_similarity(sae):
    norm_dec = torch.nn.functional.normalize(sae.W_dec, dim=1)
    dec_sims = torch.mm(norm_dec, norm_dec.T)
    triu_mask = torch.triu(
        torch.ones_like(dec_sims),
        diagonal=1,
    ).bool()
    return dec_sims[triu_mask].abs().mean()
```

*Figure 23.* Pytorch pseudocode for decoder pairwise cosine similarity

