# OpenReview forum: "Sparse but Wrong: Incorrect L0 Leads to Incorrect Features in Sparse Autoencoders"
_ICML.cc/2026/Conference — ICML 2026 regular_

### Official Review · Reviewer_Qqbk · 2026-02-23

**Soundness:** 3
**Presentation:** 3
**Significance:** 2
**Originality:** 3
**Overall Recommendation:** 5
**Confidence:** 4

**Summary:**

This paper investigates the impact of the L0 hyperparameter on the recovery of ground-truth linear features by sparse autoencoders. Studies on toy-models are used to inform a decoder cosine similarity metric for calibrating L0 choices. This metric is then applied to LLMs, however as the ground-truth features are unknown for LLMs, the choice of L0 is not validated.

**Compliance With Llm Reviewing Policy:**

Affirmed.

**Ethical Review Flag:**

Flag this paper for an ethics review.

**Key Questions For Authors:**

1. If sparse probing can be used as evidence for the correct L0, why do we need the new metric at all? Is it just avoiding hyperparameterisation and maybe some cost?

**Limitations:**

yes

**Strengths And Weaknesses:**

Soundness:
- The use of simple toy models to inform a hypothesis about features in LLMs is standard within the SAE literature (eg. Toy Models of Superposition / Towards Monosemanticity). This paper uses a satisfying range of toy models, but is perhaps somewhat too dependent on them due to a lack of ground-truth in LLMs.
- When applying this metric to choosing L0s for LLM SAEs, a comparison to a sparse probing benchmark is used to demonstrate that the decoder similarity metric method is predictive of linear interpretation quality.
- Only small (i.e. 1-2b parameter) LLMs are used. This is fairly standard in the SAE literature, but a wider range would improve confidence in these results.
- BatchTopK and JumpReLU both operate fairly similarly, and Top-K SAEs (which are also popular) aren't covered. See "Projecting Assumptions: The Duality Between Sparse Autoencoders and Concept Geometry". I think this is okay though, maybe just worth nothing somewhere.

Presentation:
- The figures and writing are clear.

Significance:
- Correctly choosing an L0 is a constant pain in training SAEs, with the advice generally being that the practitioner using the SAE should choose whatever L0 works for their purpose. This method goes some way in helping this decision making.
- This method still requires training a large sweep of SAEs, so does not fully resolve this issue.
- Some of the plots have very broad minima spanning L0s from 200 to 700 (for figure 8 LHS), and uncertainty about the question from 4.2 (Can L0 be too low and high simultaneously) limits the possible impact of this method.

Originality:
- Regularising on decoder cosine similarity is not novel, and has been suggested before eg. https://www.lesswrong.com/posts/QoR8noAB3Mp2KBA4B/do-sparse-autoencoders-find-true-features and OrtSAE

---

> ### Author Rebuttal · Authors · 2026-03-30
>
> We thank the reviewer for their thoughtful review and engagement with our work.
>
> **Why c_dec over sparse probing?**
>
> This is an excellent question. The key advantages of c_dec are:
>
> 1. **No labeled data required.** Sparse probing requires supervised probing datasets with known concept labels. c_dec is computed purely from the SAE decoder weights. It requires no labeled data, no forward passes through the LLM, and no choice of probing tasks. This is especially important as SAEs are increasingly applied beyond natural language (e.g., biology, vision), where high-quality probing datasets often do not exist at all.
> 2. **Task-agnostic.** A sparse probing evaluation measures performance on a specific set of concepts. c_dec provides a single holistic measure of dictionary quality without needing to select which concepts to probe for.
> 3. **No additional hyperparameters.** Sparse probing introduces its own hyperparameters (k in k-sparse probing, choice and number of probing tasks, dataset size). c_dec is a single scalar with no hyperparameters.
>
> We use sparse probing as a *validation* of c_dec precisely because sparse probing is an established measure of feature quality. The fact that c_dec correlates well with sparse probing performance (Figure 8) supports its validity. In domains where sparse probing datasets do not exists, such as biology of chemistry models, c_dec may be the only practical option.
>
> **Novelty of decoder cosine similarity**
>
> We appreciate the reviewer pointing out the LessWrong post and OrtSAE. We want to draw a clear distinction: those works propose *regularizing* on decoder similarity during training to encourage orthogonality. Our work uses decoder pairwise cosine similarity as a *post-hoc diagnostic metric* to guide L0 selection. These are not just different use cases; they are in direct tension. Regularizing decoder cosine similarity during training would *break* our metric, because the metric's value comes precisely from the fact that elevated c_dec is a natural signal of feature mixing. If you regularize c_dec away during training, you suppress the diagnostic signal without necessarily fixing the underlying problem (the SAE may still learn incorrect features, but with artificially orthogonalized decoder directions). Our contribution is the insight that decoder cosine similarity is *informative about L0 correctness*, not that it should be minimized as a training objective.
>
> **Broad minima**
>
> We agree that some c_dec curves have broad minima, and we appreciate the reviewer raising this. The primary value of c_dec is in identifying the clearly-wrong regions (very low L0, where c_dec spikes sharply). As Figure 8 shows, the "elbow" where c_dec begins rising steeply at low L0 is quite sharp and corresponds well to peak sparse probing performance, even when the minimum itself is broad.
>
> **Dependence on toy models**
>
> We want to note that our argument is not solely empirical. Theorem E.1 provides a formal proof that MSE loss directly incentivizes feature mixing when L0 is below the true L0, and Theorem F.1 gives a theoretical justification for the c_dec metric under more general conditions. The toy models serve to validate these theoretical predictions across increasingly realistic settings (5-feature through 50-feature), and our LLM experiments on Gemma-2-2b and Llama-3.2-1b then confirm the predictions hold in practice (Figure 8).
>
> **Top-K SAEs**
>
> We note that Top-K is not considered a state-of-the-art SAE architecture. BatchTopK was introduced specifically as an improvement over Top-K (by allowing variance in per-sample sparsity within a batch), and we are not aware of a reason to prefer Top-K over BatchTopK or JumpReLU. Since BatchTopK and Top-K are architecturally very similar (both select the top-k activations, differing only in whether selection is per-sample or per-batch), we would expect similar behavior and believe our BatchTopK results are informative for Top-K as well.

---

> > ### Author Rebuttal · Reviewer_Qqbk · 2026-03-31
> >
> > Thanks for your response and explanations.
> >
> > Wrt. c_dec vs sparse probing, I'm convinced by the generalisation property of c_dec but less so by its precision. It's nice to have a metric that might give some signal when training SAEs on non-language foundation models. My remaining concerns relate to the precision of these metrics.
> > - The sparse probing F1s are the averages of many probes, but I don't see results for individual tasks and whether they all share the average maximum.
> > - c_dec seems to poorly predict probe performance for high L0s in Figure 8 for Gemma 2 2b, though I agree that at low L0s it seems useful for obviously wrong L0s.
> >
> > On reflection, and based on the author response, I am increasing my score. I find the toy models and theory illuminating, and agree that the c_dec metric could help avoid obviously bad choices of L0 in new SAEs. I have significant reservations about the c_dec metric, and the aggregated viewpoint that this (and other related papers) use, though this paper also flags potential related concerns.

---

### Official Review · Reviewer_DBD8 · 2026-03-04

**Soundness:** 2
**Presentation:** 2
**Significance:** 1
**Originality:** 2
**Overall Recommendation:** 3
**Confidence:** 3

**Summary:**

This paper conducts analysis on the effects of sparsity constraints (L0) of SAEs, showing that overly large or small L0 both lead to mixed representations of features. Authors also challenge the soundness of reconstruction-sparisity tradeoff plot, which is usually used to evaluate the performance of an SAE architecture. Finally, the authors introduce a metric that sweeps through sparsities and finds the optimal one. Experiments are conducted on both toy models and real-world LLMs to verify the proposed arguments and metric.

**Compliance With Llm Reviewing Policy:**

Affirmed.

**Final Justification:**

I raised my score from 2 to 3 since the rebuttal solves some of my concerns and confusion. The main reason for not raising it to 4 is the paper's originality and significance. I shared similar concerns with Reviewer nLci. My concern about the similarity (or difference) of effects of "low L0" and "low width" is not fully addressed, and I think it needs more experimental justification. Authors have argued their difference in cause, whilst my point is the effects on SAE's internals from the perspective of training dynamics or final SAE features.

**Key Questions For Authors:**

Please see Weakness.

**Limitations:**

yes

**Strengths And Weaknesses:**

**Strength**
1. The discussion around the sparsity–reconstruction tradeoff is sound. The argument and designed experiments successfully show that an overly sparse SAE may instead achieve a better reconstruction.
2. The proposed metric (decoder pairwise cosine similarity) shows high consistency with sparse probing, showing its quality and predictability.

**Weakness**
1. The main argument in this paper, as has been mentioned by the authors, can be a manifestation of feature hedging. It is unclear whether the effects of having overly low L0 vs. overly low hidden dimension are fundamentally the same.
2. The phenomenon of mixed features has been discussed by [1], who also used toy experiments to demonstrate this phenomenon. The "overly low L0" can be seen as simply a manifestation of this phenomenon, thus deteriorating the novelty of the proposed argument.
3. The readability and clarity can be improved. For examples:
    a. “(afterall, it is called a sparse autoencoder)” is not rigorous enough to support “most practitioners would expect that too high an L0 will break the SAE” in Introduction.
    b. Section 3.3 is one of the main arguments; I would expect a deeper discussion and explanation around it. Authors can also consider arranging visualization results as tables/figures to provide more information on this argument.
4. Eq. 1 subtracts the input to the encoder by the decoder's bias—is this a typo?
5. Why does Figure 4 use variance explained rather than MSE? Can authors provide an MSE-sparsity plot?
6. Figure 9 (right) shows only a subtle trend of a hump, which may not support the corresponding analysis in Section 4.2. Additionally, in Section 4.2, the authors mention the trend to be gaussian, while I'm not sure if this would be a precise language to describe—do the authors just want to express that "the shape looks like a gaussian distribution"?
7. The proposed metric needs to sweep for various sparsities, which is impractical for real-world use. Furthermore, its mathematical design relies on the orthogonality of ground-truth features, which may not hold in real-world models and thus are not theoretically justified.

---

> ### Author Rebuttal · Authors · 2026-03-30
>
> We thank the reviewer for their detailed feedback.
>
> **Weakness 1: Feature hedging vs. L0 effects**
>
> This is an insightful question. Feature hedging (Chanin et al., 2025) describes what happens when SAEs have insufficient *width*: the SAE literally does not have enough latents to represent all features, so it merges correlated ones. In our work, the SAE has sufficient width but the *sparsity constraint* prevents it from using enough latents simultaneously. The key difference:
>
> - **Low width:** The SAE cannot represent all features because it lacks capacity. The solution is to increase width.
> - **Low L0 (regardless of width):** The SAE can have enough latents but is constrained from activating enough of them per input. It responds by learning latents that mix correlated features together, allowing fewer active latents to capture more information. The solution is to adjust L0.
>
> These lead to related but distinct failure modes. Furthermore, feature hedging does not study L0 as a variable, does not provide a theoretical result explaining the result due to L0 (our Theorem E.1), does not show that commonly-used sparsity-reconstruction frontier is a misleading metric, and does not propose a metric for detecting mixing due to L0. Our paper specifically characterizes the L0 dimension of this broader phenomenon and provides practical tools for practitioners.
>
> **Weakness 2: Novelty relative to [1]**
>
> [1] observes that narrow SAEs can merge correlated features. However, that work does not study the effect of L0, does not show that the sparsity-reconstruction tradeoff is misleading, does not prove that MSE loss incentivizes mixing (Theorem E.1), and does not propose a diagnostic metric for picking L0. The observation that "mixed features can happen" is a starting point, not the full story. Our paper explains a *novel case* where it happens (as a function of L0), *why* it happens (MSE loss directly incentivizes it), *how to detect it* (c_dec), and characterizes the *high-L0 failure mode* (degenerate solutions), all of which is entirely absent from prior work.
>
> **Weakness 3: Readability**
>
> We agree and will fix these in the revision. Specifically:
>
> - (a) We will replace the informal "afterall" argument with a more rigorous discussion of practitioner expectations, citing the prevalence of sparsity-reconstruction tradeoff plots in the literature.
> - (b) We will expand Section 3.3 with a more detailed discussion and include a table/figure comparing MSE values for the ground-truth vs. trained SAEs at different L0 values.
>
> **Weakness 4: Eq. 1**
>
> This is not a typo. This is the standard pre-encoder bias subtraction commonly used in SAEs, and is present even in early SAE work [1]. The encoder receives (x - b_dec) as input, where b_dec is the decoder bias. This allows the encoder to operate on centered inputs. We will add a clarifying note.
>
> **Weakness 5: Variance explained vs MSE**
>
> We used variance explained because it provides a normalized 0-1 scale that is comparable across layers (which have different activation magnitudes). We will include an MSE-sparsity plot in the revision as a supplementary figure. The qualitative conclusions are identical.
>
> **Weakness 6: Figure 9 (right)**
>
> We agree the trend in Figure 9 (right) is subtle at L0=750. We use "Gaussian" loosely to describe the shape of the distribution and will clarify this language. The key observation is that at L0=200 (near the c_dec minimum) the distribution is narrow and concentrated near 0, while at L0=10 and L0=2000 it is wider, indicating more feature mixing. At L0=750, the distribution is narrower than at L0=200 near the center but shows heavier tails (visible in the log-scale plot), which we interpret as some latents becoming more monosemantic while others mix more, consistent with Section 4.2's discussion of heterogeneous L0 preferences across latents.
>
> **Weakness 7: Practicality and orthogonality assumption**
>
> On practicality: we agree that sweeping over L0 values is expensive. However, SAE practitioners already routinely train sweeps over hyperparameters (L0/sparsity coefficient, width, learning rate) to find good configurations. The c_dec metric replaces the need for downstream evaluation, and this is especially important where we do not have sparse probing datasets (e.g. models for biology or chemistry).
>
> On orthogonality: We do not assume that features are orthogonal, as we agree that in LLMs this may not be strictly true. We only show that if the L0 is set incorrectly, feature mixing will cause the SAE decoder to become *less* orthogonal, so the ideal L0 should be when the SAE decoder is most orthogonal. The strong empirical correlation between c_dec and sparse probing performance in our LLM experiments (Figure 8) suggests the metric is robust.
>
> ### References
>
> [1] Bricken et al. "Towards Monosemanticity: Decomposing Language Models With Dictionary Learning" https://transformer-circuits.pub/2023/monosemantic-features (2023)

---

> > ### Author Rebuttal · Reviewer_DBD8 · 2026-03-31
> >
> > The author's response addresses some of my questions but does not fully solve my concerns.
> >
> > My concerns about Weaknesses 1 and 2 are solved a bit, but still remain. I understand that low width and low L0 are two different variables, but it's unclear to me whether they have the same or different internal effects on SAE features. It will be interesting to see, even empirical evidence, that they do. I encourage authors to jointly discuss them.
> >
> > Weakness 4 is still unclear to me. The [1] paper explicitly mentioned that the bias term in the encoder ($b_d$; the next equation after Eq. (1)) is **pre-encoder bias**. It is different from the decoder bias $b$ that appeared in Eq. (1). Please let me know if I misunderstood.
> >
> > Weakness 5 is not solved. The experimental result is pivotal, whilst not provided.
> >
> > [1] Bricken et al. "Towards Monosemanticity: Decomposing Language Models With Dictionary Learning"

---

> > > ### Author Response · Authors · 2026-04-01
> > >
> > > We thank the reviewer for their continued engagement.
> > >
> > > **Weaknesses 1 & 2: Joint discussion of low width vs. low L0**
> > >
> > > We appreciate this suggestion and will add a joint discussion in the revision. We expect that low width and low L0 produce similar *symptoms* (mixed features) but through different *mechanisms*: low width forces mixing because the SAE lacks latents to represent all features, while low L0 forces mixing because the SAE cannot activate enough of its existing latents simultaneously. While these are related phenomena, their manifestations are distinct. When the SAE is too narrow but has sufficient L0, SAE latents will mix components of correlated features that are not explicitly tracked by the SAE into latents that are tracked by the SAE. However, when the SAE has sufficient width but not enough L0, the SAE learns mixed latents combining features it has sufficient capacity to represent individually, because the sparsity constraint prevents it from activating enough latents simultaneously. Our toy model experiments already demonstrate this distinction: the SAEs in Figures 1-3 have sufficient width ($g = h$) yet still learn mixed features purely due to the L0 constraint. We will add a discussion of mixing due to width vs L0 in the revision.
> > >
> > > **Weakness 4: Pre-encoder bias**
> > >
> > > We believe there may be a misunderstanding. In the standard SAE formulation (as used in our paper and in Bricken et al.), the pre-encoder bias *is* the decoder bias $b_d$, reused. The full architecture from Bricken et al. is reproduced below:
> > >
> > > $\bar{x} = x - b_d$
> > >
> > > $f(x) = \text{ReLU}(W_e \cdot \bar{x} + b_e)$
> > >
> > > $\hat{x} = W_d \cdot f(x) + b_d$
> > >
> > > The same $b_d$ appears in both the pre-encoder subtraction and the decoder output. This is described explicitly in the "Advice for Training Sparse Autoencoders: Autoencoder Architecture" section of Bricken et al. (https://transformer-circuits.pub/2023/monosemantic-features#appendix-autoencoder). The encoder has its own separate bias $b_e$, but the centering term subtracted from the input before encoding is $b_d$, not an independent parameter. Our Eq. 1 is consistent with this standard formulation.
> > >
> > > **Weakness 5: MSE-sparsity plot**
> > >
> > > We have added the requested MSE-sparsity plot to our revision, available at https://anonymous.4open.science/api/repo/anon2-BF36/file/rebuttal.pdf. As expected, the qualitative conclusions are identical to the variance explained version: the trained SAE with mixed features achieves better MSE than the ground-truth SAE at every L0 below the true L0, confirming that the sparsity-reconstruction frontier is misleading regardless of whether reconstruction is measured by variance explained or MSE.
> > >
> > > ---
> > >
> > > *We hope these responses address the reviewer's remaining concerns. If so, we would kindly ask the reviewer to consider updating their score to reflect the resolution of these issues.*

---

### Official Review · Reviewer_nLci · 2026-03-10

**Soundness:** 3
**Presentation:** 4
**Significance:** 2
**Originality:** 2
**Overall Recommendation:** 2
**Confidence:** 4

**Summary:**

This paper studies how the choice of L0 affects sparse autoencoders trained on LLM activations. The main claim is that L0 is not merely a free sparsity knob in a sparsity-reconstruction tradeoff: when L0 is set incorrectly, the SAE can fail to recover disentangled underlying features and instead learn mixed or degenerate latents. To support this claim, the paper presents toy-model experiments, introduces a decoder-based proxy metric for identifying better L0 values, and evaluates the proposed ideas on SAEs trained on Gemma-2-2B and Llama-3.2-1B.

**Compliance With Llm Reviewing Policy:**

Affirmed.

**Final Justification:**

My concerns about novelty, theoretical depth, and empirical scope remain unresolved, as I elaborated in my rebuttal acknowledgment.

**Key Questions For Authors:**

I do not have major questions beyond the points raised in the Strengths and Weaknesses section. A few minor issues are listed below.

1. There is a missing period on line 132.
2. On the first line of Section 3.6, "SAES" should be corrected to "SAEs".
3. Some spacing after periods following uppercase abbreviations appears incorrect. In LaTeX, this usually requires `\@.` rather than `.` after abbreviations such as "SAE" and "LLM", otherwise the following space can be too small.

**Limitations:**

Limitations are discussed in Appendix N.

**Strengths And Weaknesses:**

**Strengths:**

1. The paper studies the selection of L0 for sparse autoencoders, which is an important and practically relevant problem in SAE training. The paper argues that an incorrect L0 can cause SAEs to fail to disentangle features, proposes a proxy metric for searching for a better L0, and supports its claims with both synthetic and LLM experiments.

2. The paper is well written and easy to follow. The research questions are clearly motivated, and the main experimental results are presented in a straightforward and accessible way.

3. The appendix contains substantial experimental detail, and the supplementary materials include source code. This makes the paper easier to inspect and improves reproducibility.

**Weaknesses:**

1. The central qualitative claim that an incorrect sparsity level can lead SAEs to learn polysemantic, correlated, or otherwise distorted features is not new. Closely related observations have already appeared in several prior works. For example, [1] explicitly states that narrow SAEs can "merge components of correlated features together, thus destroying monosemanticity." [2] reports that "low L0 SAEs tending to learn high-precision low-recall latents, and high L0 SAEs learning low-precision high-recall latents." [3] states that "lower L0 SAEs create worse sparse probes." [4] argues that "the sparsity objective incentivizes learning single latents that capture specific combinations rather than representing the underlying independent features separately." Taken together, these works already establish that sparsity strongly affects whether SAE latents are cleanly disentangled. As a result, the novelty of the paper's main argument seems limited unless the authors can better articulate what is fundamentally new beyond consolidating and repackaging these earlier observations.

2. The paper argues against using sparsity-reconstruction tradeoff plots by showing that, when L0 is below the true L0 in the toy setting, a ground-truth SAE can have worse reconstruction than a trained SAE with incorrect latents. However, this argument feels overstated. In practice, such tradeoff plots are usually examined over a sweep of L0 values, and an SAE is typically considered preferable if it achieves a better frontier over the relevant sparsity range, rather than at only one operating point. The paper therefore does show that fixed-L0 reconstruction can be misleading, but it does not fully establish that sparsity-reconstruction frontier comparisons are themselves unsound.

3. The paper appears to center its discussion around a single global "true L0" in both the synthetic and LLM experiments, which makes the framing feel somewhat incomplete. In realistic settings, different tokens, concepts, or latents may plausibly have different effective sparsity levels. This possibility is relevant to the paper's main thesis, since it weakens the idea that there should be one correct global L0 for a model or layer. Prior work has already pointed toward this broader heterogeneity [4,5], but the discussion here is limited. Section 4.2 briefly hypothesizes that some latents may effectively prefer higher or lower sparsity than others, but the paper does not develop this point in depth or study it systematically.

4. The empirical evidence is interesting, but the paper does not provide a comparably strong theoretical foundation for its broader claims. The theoretical discussion in the appendix is limited to a very simple toy setting, and much of the paper's intuition is extrapolated from that setting to LLM SAEs. In addition, the LLM experiments are limited to relatively small and older models, specifically Gemma-2-2B and Llama-3.2-1B, which makes it harder to assess how strongly the conclusions carry over to newer or larger models.

---

**References.**

[1] Chanin et al. *Feature Hedging: Correlated Features Break Narrow Sparse Autoencoders.* Mechanistic Interpretability Workshop at NeurIPS 2025.

[2] Chanin et al. *A is for Absorption: Studying Feature Splitting and Absorption in Sparse Autoencoders.* NeurIPS 2025.

[3] Kantamneni et al. *Are Sparse Autoencoders Useful? A Case Study in Sparse Probing.* ICML 2025.

[4] Bussmann et al. *Learning Multi-Level Features with Matryoshka Sparse Autoencoders.* ICML 2025.

[5] Lee et al. *Evaluating and Designing Sparse Autoencoders by Approximating Quasi-Orthogonality.* COLM 2025.

---

> ### Author Rebuttal · Authors · 2026-03-30
>
> We thank the reviewer for their thorough and carefully argued review. We appreciate the time spent engaging with the details of our work and the specific references cited.
>
> **Weakness 1: Novelty relative to prior work**
>
> We take the novelty concern seriously. None of the cited works claim that incorrect L0 corrupts the SAE, nor that there is a correct L0. We explore these in-turn:
>
> - **[1] (Feature Hedging):** Studies SAE failures due to *insufficient width*, makes no claims about SAE L0.
> - **[2] (Absorption):** Studies SAE failures due to *hierarchical features*, not L0. Observes precision-recall differences across L0 in passing, but does not explain the mechanism, nor claims that incorrect L0 can corrupt the SAE.
> - **[3] (Kantamneni et al.):** Observes that lower L0 gives worse sparse probing scores, with no explanation of why, and makes no claim that incorrect L0 can corrupt the SAE nor any claim about an optimal L0.
> - **[4] (Bussmann et al.):** Attempts to fix feature absorption due to hierarchical features in the context of Matryoshka SAEs, with no claim about optimal L0 or that incorrect L0 can corrupt the SAE.
>
> Our paper is the first to (a) systematically study the phenomenon across controlled settings, (b) prove theoretically *why* MSE incentivizes mixing when L0 is too low (Theorem E.1), (c) characterize both the low-L0 and high-L0 failure modes, (d) demonstrate that the sparsity-reconstruction frontier is misleading (Figure 4), and (e) provide a practical metric (c_dec) validated on LLMs. These prior works observed symptoms; our paper diagnoses the cause and provides a tool to address it.
>
> **Weakness 2: Sparsity-reconstruction frontier**
>
> Our argument is not limited to a single operating point. **Figure 4 shows the full Pareto frontier**, and we demonstrate that an incorrect SAE that mixes features achieves better reconstruction than the ground-truth SAE at *every point* on the frontier where L0 is below the true L0. This is a proof by counterexample that the frontier comparison itself is unreliable, not just individual operating points: following the standard methodology would lead us to discard an SAE with correct features in favor of the one with incorrect mixed features.
>
> **Weakness 3: Global vs. heterogeneous L0**
>
> We want to clarify that L0 is an *average* number of active latents over tokens across an entire dataset, not a per-token or per-latent constraint. Different tokens activate different numbers of latents, and different latents fire at different frequencies. Our claim is that there is a correct L0 (average number of active latents over a full dataset) for a given SAE architecture/settings on a given dataset, and setting it too low forces feature mixing.
>
> Section 4.2 discusses how different latents likely have different ideal firing *thresholds*, which is why JumpReLU performs better at high L0 (it adjusts thresholds per-latent). But this is separate from our main point: if the average L0 is too low, most latents will be corrupted regardless.
>
> **Weakness 4: Limited theory and small models**
>
> While we agree that extending the theory beyond the two-feature setting of Theorem E.1 would strengthen the paper, we also provide Theorem F.1, which gives a theoretical justification for the c_dec metric under more general conditions. We view these as providing the core theoretical intuition that generalizes empirically. Our toy model experiments (5-feature and 50-feature) and LLM experiments validate that the qualitative predictions of the theory hold well beyond the simplified setting.
>
> On model size: we acknowledge the limitation of using only 1-2B models. We note that SAE training at scale is computationally expensive (each sweep requires training dozens of SAEs, each on hundreds of millions of tokens), and 1-2B models are standard in the SAE literature for this reason. We have no reason to expect qualitatively different behavior at larger scales, as the feature mixing phenomenon is driven by the interaction between the sparsity constraint and the MSE loss, which is architecture-agnostic.
>
> **Minor issues:** Thank you for catching these. We will fix the missing period on line 132, correct "SAES" to "SAEs" in Section 3.6, and fix the spacing after abbreviations using `\@.` in LaTeX.

---

> > ### Author Rebuttal · Reviewer_nLci · 2026-04-04
> >
> > Thank you for the detailed rebuttal and for engaging carefully with my review. I appreciate the clarifications, but my overall assessment remains largely unchanged.
> >
> > On novelty, I believe the rebuttal does not address my central point. My claim was not that the cited papers have the same title claim or the same primary framing as this submission. Rather, my point was that the qualitative phenomenon emphasized by this paper, namely that an incorrect sparsity level leads SAEs to learn mixed, absorbed, or otherwise corrupted features, is already a common understanding in the literature. The quotes I provided in my review were intended to document exactly this point. In that sense, the observation itself does not appear sufficiently novel for a top-tier publication, even if this paper presents it under the language of a "correct L0."
> >
> > I respond to the specific novelty claims in the rebuttal below.
> >
> > > (a) systematically study the phenomenon across controlled settings
> >
> > A more systematic empirical study is useful, but in my view this is not enough for publication unless it also yields substantially new insight beyond what prior work has already established qualitatively.
> >
> > > (b) prove theoretically why MSE incentivizes mixing when L0 is too low (Theorem E.1)
> >
> > As noted in my review, Theorem E.1 is derived in a highly simplified toy setting with only two latent directions. I do not find this sufficient to support the broader claims made in the paper.
> >
> > > (c) characterize both the low-L0 and high-L0 failure modes
> >
> > I am not convinced this is new. Prior works already discuss analogous low-sparsity and high-sparsity pathologies, as reflected in the quoted passages in my review.
> >
> > > (d) demonstrate that the sparsity-reconstruction frontier is misleading (Figure 4)
> >
> > Figure 4 provides a useful counterexample showing that reconstruction at too-low L0 can favor an incorrect SAE over the ground-truth one. However, I still find the paper's broader conclusion overstated. In practice, sparsity-reconstruction plots are typically examined over a range of L0 values, and methods are compared by the overall frontier across that range. The figure therefore shows that low-L0 reconstruction can be misleading, but it does not fully establish that sparsity-reconstruction frontier comparisons are themselves unreliable in the broader sense claimed by the paper.
> >
> > > (e) provide a practical metric (c_dec) validated on LLMs
> >
> > I agree that a practical metric is useful. However, the proposed $c_\mathrm{dec}$ appears closely related in spirit to long-standing coherence or orthogonality measures in sparse dictionary learning, as well as more recent orthogonality-based viewpoints for SAEs. As such, while the metric may be helpful in this context, I do not view it as a sufficiently strong source of novelty on its own.
> >
> > My concerns regarding theory and experimental scale also remain. Theorem E.1 is limited to a very simple setting, and while the rebuttal points to Theorem F.1, that theorem serves a different purpose: it helps justify why $c_\mathrm{dec}$ can detect feature mixing, but it does not provide a comparably strong theoretical justification for the broader claim that low L0 causes feature mixing in realistic settings. On the empirical side, the experiments remain limited to relatively small and older LLMs. I understand the practical cost argument, but without at least one experiment on a larger and more recent model, I do not think the paper provides enough evidence that the reported conclusions carry over beyond the specific small-model regime studied here.
> >
> > Overall, I appreciate the authors' thoughtful response, but I do not believe the rebuttal sufficiently resolves my concerns about novelty, theoretical depth, and empirical scope.

---

> > > ### Author Response · Authors · 2026-04-04
> > >
> > > We thank the reviewer for their continued engagement. We respectfully disagree with the assessment that our novelty concerns are unresolved, and address each point below.
> > >
> > > **On novelty: the cited works do not establish what we show.**
> > >
> > > The reviewer states that "the qualitative phenomenon emphasized by this paper... is already a common understanding in the literature." We disagree. The cited works observe *symptoms* of different problems, none of which are the phenomenon we study. Specifically:
> > >
> > > - [1] (Feature Hedging) studies failures from insufficient *width*, not L0. It does not vary L0, does not claim L0 can corrupt the SAE, and does not study the mechanism we identify.
> > > - [2] (Absorption) studies failures from *hierarchical features*, a fundamentally different phenomenon with different causes, effects on the encoder/decoder, and solutions. The observation that precision/recall varies with L0 is incidental to that paper's contribution and comes with no analysis of *why*.
> > > - [3] (Sparse Probing) notes a correlation between L0 and probing performance. Noting a correlation is not the same as diagnosing a cause. The paper offers no mechanism, no theory, and makes no claim about correct L0.
> > > - [4] (Matryoshka SAEs) addresses feature absorption (again, a different phenomenon) and tangentially notes absorption worsens at low L0.
> > >
> > > None of these works claim that (a) there exists a correct L0, (b) MSE loss incentivizes feature mixing when L0 is too low, (c) the sparsity-reconstruction frontier is misleading, or (d) a diagnostic metric can identify the correct L0. Observing that "lower sparsity SAEs perform worse on some metric" in the context of studying an entirely different phenomenon does not establish the cause, the mechanism, or the practical implications. Our paper provides all of these.
> > >
> > > We also note that [1] explicitly distinguishes its failure mode (mixing with features *outside* the dictionary) from the one we study (the dictionary mixing *with itself*). These are related but distinct phenomena.
> > >
> > > **On theory: we now provide a general proof.**
> > >
> > > The reviewer noted that Theorem E.1 is limited to two latents. We have developed a general proof (Theorem E.2) extending this to SAEs with *arbitrary* numbers of latents, showing that for a bias-free tied SAE with $g \geq 2$ latents and Top-$K$ where $K < \text{true } L_0$, the disentangled solution is *not a local minimum* of expected MSE.
> > >
> > > The core intuition: when $K < \text{true } L_0$, some active features are necessarily excluded from Top-$K$ on any given input. The SAE can reduce error by tilting a firing latent $l_i$ toward an excluded feature $f_j$ via the perturbation $l_i(\epsilon) = (f_i + \epsilon f_j) / \sqrt{1+\epsilon^2}$, partially reconstructing the excluded feature "for free." This first-order gain ($dE/d\epsilon \mid_{\epsilon=0} = -2 m_i m_j < 0$) is strictly negative, while the costs from over-reconstruction (when both fire) or spurious activation (when the mixed feature is absent) are only $O(\epsilon^2)$. Combining across events gives $dL/d\epsilon \mid_{\epsilon=0} = -2 P(A) \cdot E[m_i m_j \mid A] < 0$, so the net gradient always points away from the disentangled solution.
> > >
> > > Crucially, this applies to *every* pair of features that co-occur while one is excluded by Top-$K$, which is most pairs when $K < \text{true } L_0$, explaining why latents become corrupted (Figure 1). A full writeup will be added as Appendix E.2.
> > >
> > > **On the sparsity-reconstruction frontier.**
> > >
> > > The reviewer states that practitioners compare methods by "the overall frontier across that range." **This is precisely what Figure 4 shows**: the trained SAE with incorrect features achieves equal or better reconstruction than the ground-truth SAE at *every point* on the frontier, strictly dominating wherever L0 < true L0. The entire frontier is Pareto-superior. Following the standard methodology of preferring the better frontier would lead practitioners to discard the correct SAE. This is not a single-point comparison, it demonstrates that the frontier comparison itself is unreliable.
> > >
> > > **On c_dec and existing coherence measures.**
> > >
> > > We want to be precise: we do not claim that measuring cosine similarity between dictionary elements is novel in isolation. Our contribution is the *insight* that c_dec is diagnostic of L0 correctness: it spikes when L0 is wrong and is minimized at the correct L0. Prior work on dictionary coherence (e.g., OrtSAE) proposes *regularizing* c_dec during training, which would actually *suppress* the diagnostic signal. Using c_dec as a post-hoc diagnostic for L0 selection is a distinct and, to our knowledge, new contribution.
> > >
> > > **On model scale.**
> > >
> > > We acknowledge this limitation. SAE training at scale requires training dozens of SAEs on hundreds of millions of tokens. Our theoretical results (Theorems E.1 and E.2) are scale-agnostic: the MSE pressure to mix when L0 < true L0 depends on the interaction between sparsity and the loss, not model size.

---

### Official Review · Reviewer_f8ar · 2026-03-12

**Soundness:** 3
**Presentation:** 3
**Significance:** 3
**Originality:** 2
**Overall Recommendation:** 4
**Confidence:** 3

**Summary:**

This paper investigates how the sparsity L0 hyperparameter affects the quality of features learned by SAEs. From toy model experiments and LLM experiments on Gemma-2-2b and Llama-3.2-1b, the authors demonstrate that when L0 is set too low, SAEs mix correlated features together rather than learning disentangled, monosemantic latents. Therefore, this behaviour actually improves reconstruction loss, meaning standard sparsity–reconstruction tradeoff plots are misleading. When L0 is too high, the SAE also finds degenerate solutions. The authors propose a proxy metric, decoder pairwise cosine similarity (c_dec), to help practitioners detect when L0 is set incorrectly, and show it correlates with peak sparse probing performance on real LLMs.

**Compliance With Llm Reviewing Policy:**

Affirmed.

**Final Justification:**

The rebuttal addressed my questions, I believe my original score is a fair assessment of this work, which is more of an analysis paper. I don't have any strong opinions regarding the originality of this work, and wouldn't mind being accepted if other reviewers champion it.

**Key Questions For Authors:**

Q1. How does you paper relates to dense features described in the paper "Dense SAE Latents Are Features, Not Bugs"?

Q2. What are some of the implications for steering and model editing when L0 is incorrect?

**Limitations:**

Yes

**Strengths And Weaknesses:**

Strengths

S1. Clear motivation and important core findings. The field (prior to this paper) has largely treated the sparsity–reconstruction tradeoff as the primary evaluation axis for SAEs, and this paper provides a compelling argument that this practice is fundamentally flawed.

S2 Experiments from small (5-feature) to large (50-feature) toy models is well-structured. The insight that negative correlations is also important for SAE researchers, because semantically unrelated features will have negative correlation components mixed in. The paper also trains extensive sweeps of SAEs on Gemma-2-2b and Llama-3.2-1b and validates that the proposed c_dec metric's elbow corresponds to peak k-sparse probing performance, which is an improvement over the original version

S3. Theorem E.1  provides a clean proof that MSE loss directly incentivizes feature mixing when L0 is constrained below the true L0, using a two-feature model.

Weaknesses

W1. The c_dec metric has notable limitations that are somewhat undersold. While the authors acknowledge in Section 6 that c_dec is not a perfect guide, the practical limitations are more significant than the discussion suggests.

W2. Missing analysis of relationship between width and L0. All experiments use a fixed SAE width, while the "correct L0"  can depends on the dictionary size, and whether wider SAEs are more or less sensitive to L0 misspecification.

W3. The paper could better address the training dynamics question. Appendix H shows that transitioning from too-low L0 to the correct L0 fails to recover correct features, while transitioning from too-high to correct succeeds. This is a very interesting finding with practical implications (it suggests starting with high L0 and annealing down), this can be merged into the main text if accepted.

---

> ### Author Rebuttal · Authors · 2026-03-30
>
> We thank the reviewer for their positive assessment and constructive suggestions. We are glad the core findings, experimental range, and Theorem E.1 were appreciated.
>
> **W1: c_dec limitations**
>
> This is a fair point, and we will be more upfront about limitations in the revision. Specifically: (1) the broad minima in some c_dec curves mean the metric is better at identifying the clearly-wrong low-L0 regime than at pinpointing a precise optimal L0, and (2) while c_dec correlates well with sparse probing (Figure 8), the correlation is not perfect, and we do not recommend using c_dec as the sole criterion for L0 selection if other proxy metrics are available.
>
> **W2: Width-L0 interaction**
>
> This is an excellent point. All our experiments use a fixed width, and the "correct L0" will likely shift as width changes and the SAE captures different numbers of features with different firing frequencies. Conceptually, we would expect narrower SAEs to have a lower ideal L0 than wider SAEs on the same dataset. We will add a discussion of this in the revision.
>
> **W3: Training dynamics (Appendix H)**
>
> We agree this is a practically valuable finding and will promote the key result to the main text in the revision. The asymmetry (that annealing from high L0 succeeds but annealing from low L0 fails) has direct practical implications: practitioners should err on the side of starting with higher L0 if they plan to anneal. This connects to our theoretical results: at low L0, the mixed latents represent a *better* MSE solution (Section 3.3), creating a local minimum that gradient descent cannot easily escape even when the L0 constraint is relaxed. We note that some practitioners already anneal from high to low L0 during training [1,2], although we suspect they do not understand the theoretical reasons why it is beneficial.
>
> **Q1: Relationship to "Dense SAE Latents Are Features, Not Bugs"**
>
> Great question. That work argues that high-frequency SAE latents perform important roles and should not be forced to be sparse. While we do not explicitly test out some of the phenomenon in that paper (e.g. anti-podal latents), we expect that forcing L0 too low will cause the SAE to mix those features into sparser proxy latents. Our c_dec metric could potentially help identify the L0 at which dense features begin to be suppressed. More broadly, both papers argue against the assumption that sparser is always better.
>
> **Q2: Implications for steering and model editing**
>
> If L0 is too low and SAE latents mix correlated features, then steering on a single SAE latent would activate a huge mixture of underlying concepts rather than a single clean concept. Likely this mixing just breaks the LLM, as many correlated concepts can never actually all co-occur together (e.g. an animal can't be a dog and a cat and a bird and a fish at the same time). Even worse, since we expect negative components of anti-correlated features to be mixed together, we could easily be steering in directions that are completely nonsensical to the LLM, as there is no guarantee that the negative of each feature direction is itself a valid concept. This could explain some of the unpredictable results reported in SAE-based steering work, as we expect this steering is pushing the LLM out-of-distribution by steering in nonsensical directions.
>
> ### References
>
> [1] Conerly, T et al. "Circuits Update - January 2025" https://transformer-circuits.pub/2025/january-update/index.html#DL (2025)
>
> [2] He, Zhengfu, et al. "Llama scope: Extracting millions of features from llama-3.1-8b with sparse autoencoders." arXiv preprint arXiv:2410.20526 (2024).

---

> > ### Author Rebuttal · Reviewer_f8ar · 2026-04-03
> >
> > Thanks for addressing my questions.

---

### Decision · Program_Chairs · 2026-04-30

**Decision:**

Accept (regular)

**Comment:**

This paper studies the effect of the L0 sparsity hyperparameter on Sparse Autoencoders (SAEs), and asks whether L0 is merely a free sparsity knob or whether it must be set correctly for SAEs to recover disentangled features. The reviewers generally agreed that this is a practically important question, and found the paper clear, well motivated, and supported by a useful combination of toy-model analysis and LLM experiments. They found the central empirical message convincing: if L0 is set too low, SAEs can learn mixed features rather than cleanly disentangled ones; if L0 is set too high, SAEs can also exhibit degenerate feature-mixing behavior. They also viewed the proposed decoder cosine similarity metric as a potentially useful guide for detecting obviously bad L0 choices, especially in settings where labeled probing benchmarks are unavailable. At the same time, the reviewers raised several concerns. They questioned how novel the main qualitative phenomenon is relative to recent work on feature hedging, absorption, and sparse probing, and some felt that the paper is stronger as a careful diagnosis and synthesis of known effects than as a fundamentally new discovery. They also noted that the theory remains limited to simplified toy settings, while the LLM experiments are restricted to relatively small models, making the broader generality of the conclusions harder to assess. Further concerns focused on the interaction between L0 and width, the limited precision of the proposed metric when minima are broad, and the fact that the method is more helpful for ruling out clearly wrong low-L0 regimes than for identifying a uniquely correct value. Overall, the reviewers viewed the paper as a clear and useful contribution to SAE practice, but were more cautious about its novelty and about how strongly its claims extend beyond the settings studied here.